# Spatial variation and predictors of composite index of HIV/AIDS knowledge, attitude and behaviours among Ethiopian women: A spatial and multilevel analyses of the 2016 Demographic Health Survey

**Aklilu Habte**[1]*, **Habtamu Mellie Bizuayehu**[2], **Yosef Haile**[3], **Daniel Niguse Mamo**[4], **Yordanos Sisay Asgedom**[5]

**1** School of Public Health, College of Medicine and Health Sciences, Wachemo University, Hosanna, Ethiopia, **2** School of Public Health, Faculty of Medicine, The University of Queensland, Brisbane, Australia, **3** Department of Public Health, School of Medicine, College of Health Science, Arba Minch University, Arba Minch, Ethiopia, **4** Department of Health Informatics, School of Public Health, College of Medicine and Health Sciences, Arba Minch University, Arba Minch, Ethiopia, **5** Department of Epidemiology, College of Health Science and Medicine, Wolaita Sodo University, Wolaita Sodo, Ethiopia

* akliluhabte57@gmail.com

**Data Availability Statement:** The data for this study were obtained from the DHS program with a

## Abstract

### Background

Although the dissemination of health information is one of the pillars of HIV prevention efforts in Ethiopia, a large segment of women in the country still lack adequate HIV/AIDS knowledge, attitude, and behaviours. Despite many studies being conducted in Ethiopia, they mostly focus on the level of women's knowledge about HIV/AIDS, failing to examine composite index of knowledge, attitude, and behaviour (KAB) domains comprehensively. In addition, the previous studies overlooked individual and community-level, and spatial predictors. Hence, this study aimed to estimate the prevalence, geographical variation (Hotspots), spatial predictors, and multilevel correlates of inadequate HIV/AIDS-Knowledge, Attitude, and Behaviour (HIV/AIDS-KAB) among Ethiopian women.

### Methods

The study conducted using the 2016 Ethiopian Demographic and Health Survey data, included 12,672 women of reproductive age group (15–49 years). A stratified, two-stage cluster sampling technique was used; a random selection of enumeration areas (clusters) followed by selecting households per cluster. Composite index of HIV/AIDS-KAB was assessed using 11 items encompassing HIV/AIDS prevention, transmission, and misconceptions. Spatial analysis was carried out using Arc-GIS version 10.7 and SaTScan version 9.6 statistical software. Spatial autocorrelation (Moran's I) was used to determine the non-randomness of the spatial variation in inadequate knowledge about HIV/AIDS. Multilevel

reasonable request. Thus, the one who needs the data supporting the findings of this study can get it in anonymised form from the DHS website at https://dhsprogram.com/Countries/ upon reasonable request in the same manner as the authors did.

**Funding:** The author(s) received no specific funding for this work.

**Competing interests:** The authors have declared that no competing interests exist

**Abbreviations:** AOR, Adjusted odds ratio; AIC, Akaike's Information Criterion; BIC, Bayesian Information Criterion; EDHS, Ethiopian Demographic Health Survey; HIV/AIDS, Human Immunodeficiency Virus Acquired Immunodeficiency Syndrome; ICC, Intraclass Correlation Coefficient; KAB, Knowledge, Attitude, and Behaviour; PCV, Proportionate Change in Variance; SSA, Sub-Saharan African; WHO, World Health Organization.

multivariable logistic regression was performed, with the measure of association reported using adjusted odds ratio (AOR) with its corresponding 95% CI.

## Results

The prevalence of inadequate HIV/AIDS-KAB among Ethiopian women was 48.9% (95% CI: 48.1, 49.8), with significant spatial variations across regions (global Moran's I = 0.64, p<0.001). Ten most likely significant SaTScan clusters were identified with a high proportion of women with inadequate KAB. Somali and most parts of Afar regions were identified as hot spots for women with inadequate HIV/AIDS-KAB. Higher odds of inadequate HIV/AIDS-KAB was observed among women living in the poorest wealth quintile (AOR = 1.63; 95% CI: 1.21, 2.18), rural residents (AOR = 1.62; 95% CI: 1.18, 2.22), having no formal education (AOR = 2.66; 95% CI: 2.04, 3.48), non-autonomous (AOR = 1.71; 95% CI: (1.43, 2.28), never listen to radio (AOR = 1.56; 95% CI: (1.02, 2.39), never watched television (AOR = 1.50; 95% CI: 1.17, 1.92), not having a mobile phone (AOR = 1.45; 95% CI: 1.27, 1.88), and not visiting health facilities (AOR = 1.46; 95% CI: 1.28, 1.72).

## Conclusion

The level of inadequate HIV/AIDS-KAB in Ethiopia was high, with significant spatial variation across regions, and Somali, and Afar regions contributed much to this high prevalence. Thus, the government should work on integrating HIV/AIDS education and prevention efforts with existing reproductive health services, regular monitoring and evaluation, and collaboration and partnership to tackle this gap. Stakeholders in the health sector should strengthen their efforts to provide tailored health education, and information campaigns with an emphasis on women who lack formal education, live in rural areas, and poorest wealth quintile should be key measures to enhancing knowledge. enhanced effort is needed to increase women's autonomy to empower women to access HIV/AIDS information. The media agencies could prioritise the dissemination of culturally sensitive HIV/AIDS information to women of reproductive age. The identified hot spots with relatively poor knowledge of HIV/AIDS should be targeted during resource allocation and interventions.

## Introduction

Human immunodeficiency virus (HIV) and Acquired immunodeficiency syndrome (AIDS) remain a global public health issue, with 40.4 million deaths and 39.0 million individuals living with HIV in 2022; Africa was responsible for two-thirds (25.6 million) of infections worldwide [1]. In Ethiopia, adult (15–49) HIV prevalence was 0.93% in 2019, with higher infections among women compared to men (1.22% vs 0.64%). There were broad geographical variations of HIV infections in Ethiopia, with the highest prevalence in Gambella (4.8%) and Addis Ababa (3.42%) and the lowest in Somali region (0.01%) [2]. Thus, Sustainable Development Goal 3 (SDG3) focuses on hindering HIV/AIDS spread, which necessitates not only fostering for equipping the population with evidence-based knowledge about HIV/AIDS but also ensuring equity of information access across various geographic areas [3, 4]. Achieving comprehensive HIV/AIDS knowledge, attitude, and beahviour at the population level is among the well-

known strategies for HIV/AIDS prevention [5, 6] through reducing HIV-related stigma, and encouraging testing and seeking treatment [7].

Globally, less than 30% of women of reproductive age have an in-depth knowledge of HIV/AIDS [8]. African women have also limited knowledge, attitude, and behaviour regarding how HIV is transmitted, the repercussions of infection, and methods of prevention [9–12]. On the other hand, new HIV infections among them are increasing, due to a lack of comprehensive knowledge, unfavourable attitude, and poor preventive behaviours regarding its transmission and prevention [13, 14]. Thus, to reduce the prevalence, women of reproductive age need to have adequate information regarding HIV transmission, prevention, and misconceptions [14–16].

One of the pillars of HIV prevention efforts in Ethiopia has been the dissemination of information about HIV transmission to avoid engaging in risky behaviours that enhance HIV transmission [17, 18]. However, a significant proportion of women in the country still lack adequate knowledge, attitude, and preventive behaviour about HIV/AIDS [17, 19, 20]. Socio-demographic and economic factors (age, marital status, literacy, wealth index), region, residence, media exposure, history of HIV test testing and counselling, age at marriage, and history of multiple sexual encounters were all known to influence the level of comprehensive knowledge [17, 19, 21].

Despite many studies being conducted in Ethiopia, they mostly focus on the level of women's knowledge about HIV/AIDS [19, 21–23], failing to assess the KAB domains comprehensively. Furthermore, they were mainly focused on the prevention of mother-to-child transmission and did not address individual, community-level, and spatial predictors. This study addresses those gaps by adopting a more holistic approach in terms of outcome measurement (using a comprehensive set of items) and ways of data analysis (using both spatial and multilevel analyses). Moreover, it also considered attitude-related content in terms of some misconceptions surrounding HIV/AIDS, which were previously overlooked. Hence, this study aimed to estimate the prevalence, geographical variation (hotspots), spatial predictors, and multilevel (individual and contextual) correlates of HIV/AIDS-KAB among Ethiopian women.

HIV/AIDS is one of the top priorities of the public health agenda in Ethiopia and there are efforts to enhance knowledge at various administrative units. However, there is limited evidence regarding the regions and locations that performed well or limited achievement, which will be addressed by the spatial analysis in this study. Thus, the evidence from the study might be important to resource prioritisation and tailored interventions [24]. The findings will also help to understand individual and community-level predictors of comprehensive HIV/AIDS KAB, offering insights that can inform targeted interventions, policy decisions, and community empowerment, and ultimately contribute to HIV/AIDS prevention and equity. Furthermore, the findings of this study have value as they shed light to the scientific community on the levels, geographical distribution, and causes of poor comprehensive HIV/AIDS-KAB among Ethiopian women. This evidence-based approach is essential for advancing public health efforts and addressing the possible challenges faced by women in Ethiopia in the prevention of HIV/AIDS.

## Methods

### Study area, period, and data source

This cross-sectional study was based on an analysis of the women (IR) dataset of the 2016 Ethiopian Demographic and Health Survey (EDHS) report; a nationally representative survey conducted by the Central Statistics Agency (CSA) from January 18, 2016 to June 27, 2016 [25]. Ethiopia, an East African country, is divided administratively into nine regional states (Tigray,

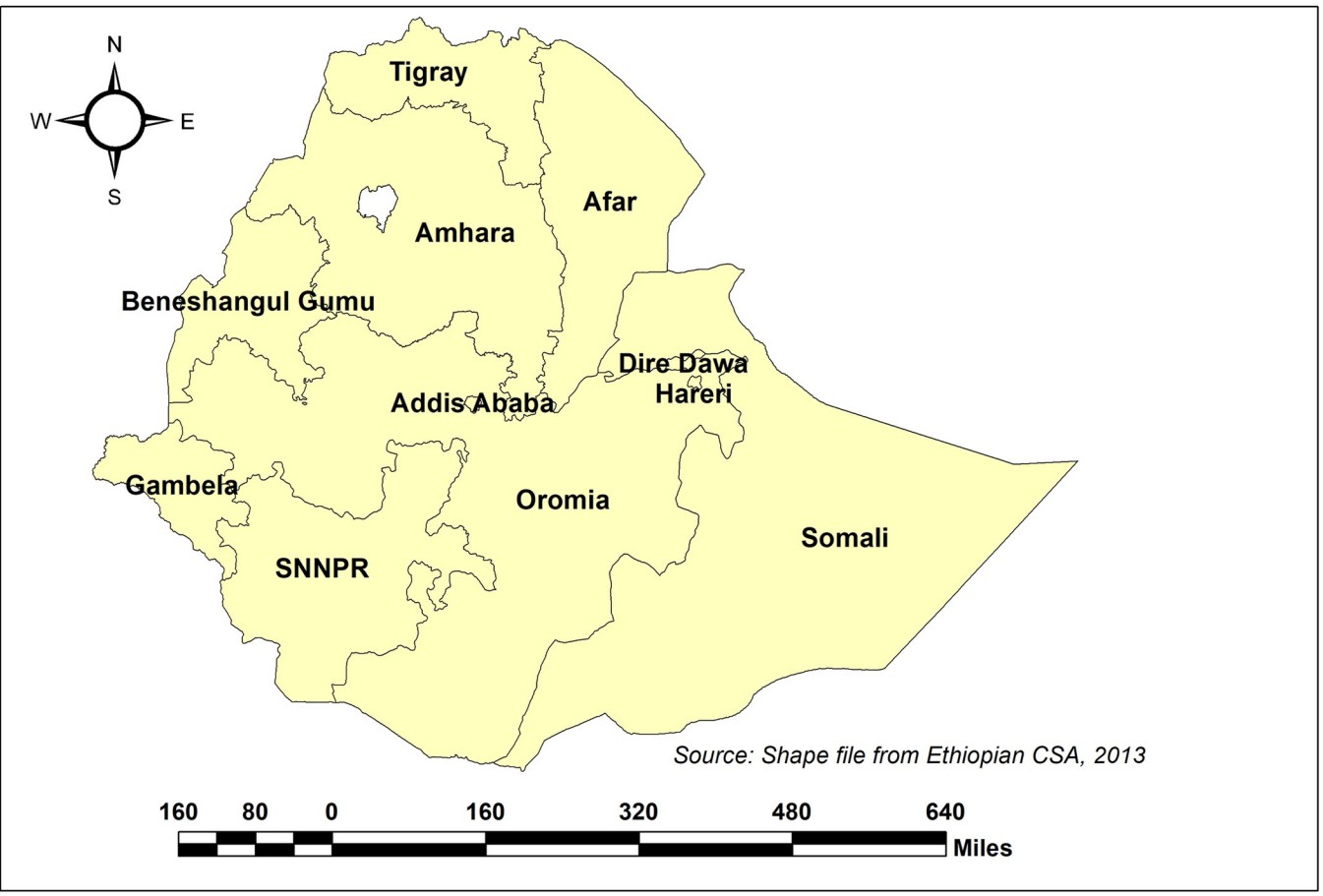

**Fig 1. Administrative map of Ethiopia, Ethiopian CSA, 2013.**

Afar, Amhara, Oromia, Somali, Benishangul-Gumuz, Southern Nations, Nationalities, and People's Region (SNNPR), Gambella, and Harari) and two administrative cities (Addis Ababa and Dire-Dawa) [25] (**Fig 1**).

## Population of the study and eligibility criteria

The source population comprised all reproductive women (15–49 years). The study population on the other hand were those women from the selected Enumeration areas (EAs), who had complete information on the knowledge assessment questions and other relevant variables. Women who have missing data about their residential area locations via geographical positioning system (GPS) were excluded from the spatial analysis.

## Sampling procedure and data collection tools

A stratified two-stage cluster sampling technique was used to select study participants. The survey used the enumeration areas (EAs) from the 2007 Ethiopian population and housing census (PHC) as the primary sampling unit (PSU) and households as the secondary sampling unit (SSU). First, 645 EAs (202 and 443 from urban and rural areas, respectively) were picked using a probability proportional to EA size (PPS). Then, in each of the selected EAs, a household listing operation was run from September to December 2015, and the resulting lists were used as a sampling frame for the selection of households. The newly formed household listing

was then used to select a fixed number of 28 households per cluster with an equal probability of systematic selection [25]. Each survey cluster's geographic coordinates were obtained using the spatial data of 2016 EDHS. Data were collected by using the Woman's Questionnaire, which contains background characteristics, maternal health service, and health system-related characteristics.

## Measurement of variables of the study

The outcome variable was the level of women's HIV/AIDS knowledge attitude and behaviour, defined as the understanding of modes of transmission, means of prevention, and misconceptions. Composite index of HIV/AIDS-KAB was assessed by 11 questions that dealt with modes of transmission, misconceptions, and means of prevention (Table 3). Sample questions were: Can HIV be prevented by limiting sex to one faithful uninfected partner?, can a person get HIV from mosquito bites?, can using condoms prevent HIV transmission?. . . The answers were coded as (1 = correct response, and 0 = incorrect one). A composite index of HIV.AIDS-KAB that ranges from 1 to 11 was created, and it was then dichotomized into inadequate knowledge (< mean score = 1) and adequate knowledge (≥mean score = 0).

**Explanatory variables.** Following a review of relevant literature [11, 19, 26–28], individual and community-level determinants of comprehensive knowledge about HIV were identified. Individual-level characteristics included socio-demographic, obstetric, and healthcare-related aspects that were unique to each woman, and community-level factors, such as residence and region, were shared by all women residing in the same community (cluster) (Table 1).

## Data management and statistical analysis

Statistical software such as STATA version 14, ArcGIS version 10.7, and SaTScan version 9.6 were used to analyse the results of this study. A weighting factor $\left(\frac{v005}{1000000}\right)$ was used to ensure the representativeness of the data which results in robust statistical estimations [31]. Descriptive statistics such as frequencies and percentages were computed.

**Spatial analyses.** To execute the spatial analysis, the weighted proportions of outcome and explanatory variables were computed in STATA and the spatial analysis was computed using ArcGIS version 10.7, and SaTScan version 9.6.

**Spatial and incremental autocorrelation.** The global spatial autocorrelation was performed using Moran's index to determine whether inadequate HIV/AIDS-KAB among women was dispersed, clustered, or randomly distributed. Moran's, I value was considered statistically significant at p<0.05 indicating geographical HIV/AIDS knowledge variations (clustered) [32, 33]. In addition, incremental autocorrelation was employed to estimate the average nearest neighbour, along with the minimum and maximum distance bands and their accompanying z-scores. Z-scores measure the strength of spatial clustering, and statistically significant peak z-scores indicate locations where spatial mechanisms favouring clustering are most evident. These peak distances are often appropriate values to use for tools with a distance band or distance radius parameter (S1 File).

**Hot spot analysis (Getis-Ord Gi\* statistics).** Getis-Ord Gi\* statistics were performed to identify significant hot spot and cold spot areas for inadequate comprehensive knowledge about HIV transmission and prevention. The Z-score and p-value were determined to figure out the statistical significance of clustering; a high and low Gi\* denote a "hot spot," and "cold spot." respectively [34, 35].

**Spatial interpolation.** The spatial interpolation technique was used to estimate the proportion of women in the country with inadequate HIV/AIDS-KAB in un-sampled areas, based

**Table 1. List of potential predictors of HIV/AID-KAB extracted from the EDHS 2016 report.**

**Individual-level factors**

| Variables | Description and categorisation |
|---|---|
| Age | Women's age at the time of the survey and categorised as 15–19, 20–34, and 35–49 years |
| Marital status | Current status of marriage or cohabitation, categorised as cohabited, never in union, and non-marital relation |
| Educational status | Categorised as having no formal education, primary, secondary, and higher education |
| Family size | Number of household members at the time of data collection and dichotomised as ≤5 and >5 |
| Wealth index | Calculated using easy-to-collect data on a household's ownership of selected assets, such as televisions and bicycles; materials used for housing construction; and types of water access and sanitation facilities. Finally, it was categorised into quintiles; poorest, poorer, middle, richer, and richest |
| Parity | The number of living children the woman had at the time of the survey, and was grouped as nulliparous, primiparous, multiparous, and grand multiparous |
| Contraceptive uptake | The proportion of women who ever used any contraceptive and categorised as user and non-user |
| Recent sexual activity | The proportion of women about their recent sexual activity and the responses were categorised as never had sex, not active in the last 4wk (due to postpartum or not), and active in the last 4 weeks |
| Difficulty in accessing healthcare | Percentage of women who reported that they have serious problems in accessing health care for themselves when they are sick, by type of problem: Getting permission to go for treatment, getting money for treatment, and ease of distance to the health facility. The responses were categorised as 'Not big problem' or 'Big problem' |
| Covered by health insurance | Percentage of women and men ages 15–49 covered by any health insurance schemes. |
| Media exposure | Number of women who are exposed to specific media with various frequencies: read a newspaper, watch television, and listen to radio. The responses were categorised as Not at all, less than once a week, and at least once a week |
| Autonomy in decision-making [*] | Responses were categorized into low, middle, and high |

**Community-level factors**

| | |
|---|---|
| Residence | The area where respondents lived and categorised as urban and rural |
| Region | The geographically delineated area where the woman was resided at the time of the survey. Three categories were created as Small periphery regions (Afar, Somali, Benishangul, and Gambella), Major central regions (SNNPRs, Tigray, Amhara, and Oromia), and Metropolitans (Addis Ababa, Dire Dawa, and the Harari region) |
| Community poverty | Was created from household wealth index by recoding poorest as a poor, and then, the proportion of women from households with poor household wealth index was calculated and categorized as low poverty level (those with 50%) and higher poverty level (those with <50%) using national median value. |
| Community women illiteracy | Community-level education was created by aggregating the individual level woman's education at cluster level by taking the proportion of women with no education, which was similar as performed for the wealth index. Then we categorized it as a low and higher level of community education using a national median value like media exposure. |
| Community media exposure | Was created from the women's exposure to radio, newspaper/magazine, and television (after merging these variables and recoding into Yes and No). |
| | Then, the proportion of women who had exposure to at least one of these media and categorized were into low (if <50% of women had exposure to at least one media) and high (if 50% of women had exposure to at least one media) community-level media exposure |

*Autonomy in decision-making: was assessed by compiling and categorising replies to three questions about who takes the final decision for the family on purchasing, visiting to relatives, and seeking health care. The response categories were (i) woman alone, (ii) woman and husband/partner, (iii) husband/partner alone, (iv) someone else, and (v) others. For each question, responses (i) or (ii) got a score of 1, indicating good decision-making capacity, whereas the remaining responses received a score of 0, indicating weak decision-making capacity. Each of the three components' responses were summed together to yield an overall score ranging from 0 to 3. Finally, a composite score was divided into two distinct groups: low and high for "0 to 2" and "3" scores [29, 30].

on observed measurements in the sampled EAs. Although there are different deterministic and geostatistical interpolation methods available, we opted to use the Ordinary Kriging spatial interpolation method since it incorporates spatial autocorrelation and statistically optimises the weight, and have low residual and mean square error [36, 37].

**The spatial scan statistical (SaTScan) analysis.** Significant hotspots (spatial clusters) with inadequate comprehensive HIV/AIDS-KAB were estimated using the Bernoulli model from the SaTScan analysis. When using the Bernoulli model, women with inadequate knowledge were classified as cases, otherwise as controls. To determine the number of observed and

expected observations inside the window at each point, SaTScan statistics scanned the space progressively. By using the default maximum spatial cluster size of <50% of the population as an upper limit, it was possible to detect both small and large clusters and ignore clusters containing more than the maximum limit [38]. A likelihood ratio (LLR) test statistic with the corresponding p-value was used to identify and rank the clusters in which the observed proportion of inadequate comprehensive knowledge was significantly higher than the expected one or not [39].

**Spatial regression.** Global (ordinary least squares) and local (geographically weighted regression) techniques are used in spatial regression analysis [40, 41].

Ordinary Least Squares (OLS) regression is a global (model that predicts only one coefficient per independent variable over the whole study area [41]. Assumptions such as the statistical significance of the intercept and predictors, and the presence of multicollinearity among explanatory variables were checked. Variables having multicollinearity (variance inflation factors (VIF>10)) were removed from the model iteratively [40, 42]. Furthermore, The Koenker statistics in the model, revealed a statistically significant p-value (p<0.001), indicating that the global regression model is inconsistent across the study area, suggesting that the local (GWR) model is required to appropriately estimate the model parameters. Finally, a model comparison between global and local regression analysis was done, and the local model outperformed the global one on the basis of increase in adjusted R2 and decrease in Akaike's Information Criterion (AIC) (Tables 4 and 5).

**Multilevel regression model.** We employed multilevel modelling due to the DHS data were hierarchical, and as the outcome variable was binary, we used multilevel multivariable logistic regression analysis.

A multilevel bivariable logistic regression analysis was performed to look into the independent association between knowledge and explanatory variables, with variables with p< 0.25 included in multivariable analysis. Statistically significant factors (at p< 0.05) along with their adjusted odds ratio(AOR) were reported. At a cutoff point of 10, the Variance Inflation Factor (VIF) was used to detect multicollinearity among covariates and found none (the VIF ranged from 1.01 to 1.97 with a mean of 1.74).

**Random effects (measures of variation).** Four separate models were fitted: the first (model 1) was without any covariate (empty model), the second (model 2) merely had individual-level factors, the third (model 3) with only community-level variables, and the fourth (Model 4/full model) with both individual- and community-level variables. For measures of variation, the intra-class correlation coefficient (ICC) and proportional change in variance (PCV) metrics were computed.

ICC is a measure of the degree of heterogeneity of having inadequate HIV/AIDS-KAB between clusters, and it was determined using: $ICC = \frac{var(b)}{Var(b)+Var(w)}$, where Var(b) is the variance at the group level and Var(w) is the predicted individual variance component, which is $\pi^2/3$ ≈3.29 [43].

Proportional Change in Variance (PCV) was estimated as

$PCV = \frac{(Va-Vb)}{Va} * 100$, where $V_a$ is the variance of the initial model (null model) and $V_b$ = variance of the subsequent models (models 2, 3, and 4).

MOR measured the unexplained heterogeneity in the odds of inadequate comprehensive HIV/AIDS knowledge from one cluster to another one and is determined by using [44]:

$$MOR = \frac{\exp(\sqrt{2*Vb}}{2a} * 0.6745 \approx \exp\left(\sqrt{2*Vb}\right)$$

Following the comparison, the fourth model with the lowest deviance, and Akaike's Information Criterion (AIC) values was found to be the best-fit model for this study (Table 6).

## Ethical consideration and consent to participate

All methods and processes were carried out per the relevant rules and regulations of the Helsinki Declaration. Following registration, the DHS office issued written permission to utilise the DHS datasets. Furthermore, as this is secondary data, the ethics committee of Wachemo University College of Medicine and Health Sciences declared that no formal ethics approval was required, with a written letter at the reference number (WCU/341/2023).

## Results

### Background characteristics of the respondents

This study considered a weighted sample of 12,672 women with a mean (±SD) age of 27.77 (±9.11) years (Table 2). The vast majority of respondents (87.8% and 74.8%, respectively) were from large central regions and rural residents. Almost two-thirds (64.0%) of women have cohabited marital status, and one-third (3,854(30.4%)) were found in the richest wealth quintile. Inadequate comprehensive HIV/AIDS-KAB was observed among 5,514 (58.2%) of rural residents, 1,229(66.8%) of women in the poorest wealth quintile, and 3,472(64.3%) of women lacking formal education. Three out of every ten women (30.8%) were multiparous (had 2–4 living children), and nearly three-quarters (72.8%) were not using contraception. The vast majority of women (82.0%) had higher decision-making autonomy, and more than half (55.5%) had visited health facilities in the last 12 months. Comprehensive HIV/AIDS-KAB was low among women who are grand multiparous (62.0%), non-autonomous (70.8%), and who never watched television (59.3%) (Table 2).

### The level of inadequate HIV/AIDS-KAB and regional disparities

About half of women (48.9%: 95% CI: 48.1, 49.8) were found to have inadequate HIV/AIDS-KAB. The highest and the lowest proportion of women with inadequate KAB were reported in Somali (75.4%) and Addis Ababa (15.2%) respectively (Fig 2). Most women have awareness about the transmission of HIV during breastfeeding (91.7%), pregnancy (80.6%), and delivery (86.8%) (Table 3).

### Results of spatial analysis

**Spatial distribution of inadequate knowledge.** Somali, the western borders of Afar, Benishangul Gumuz, and Gambella had a higher proportion of women with inadequate HIV/AIDS-KAB. In contrast, Tigray, Addis Ababa, and the eastern border of Amhara showed good KAB (Fig 3).

**Spatial and incremental autocorrelation of inadequate HIV/AIDS-KAB.** The global spatial autocorrelation analysis demonstrated that the geographical distribution of inadequate HIV/AIDS-KAB across the country was non-random (i.e. there was significant spatial variation) (global Moran's I = 0.64, p<0.001). The clustered patterns (on the right sides) suggest that there was a high proportion of inadequate KAB throughout the study area (Fig 4).

The incremental spatial autocorrelation across a series of distances was displayed by a line graph with a corresponding z-score to establish the average nearest neighbour and minimum and maximum distance band. A total of ten distance bands were uncovered with an initial distance of 159213.0 meters, with the first maximum peak (clustering) identified at 234495.4 meters (S1 File).

**Table 2. Distribution of characteristics of women: Inadequate HIV/AIDS-KAB and bivariable analysis, EDHS 2016.**

| Variable categories | Total Weighted frequency (%) [N = 12,672] | Inadequate HIV/AIDS-KAB (%) | COR[95% CI] | p-value |
|---|---|---|---|---|
| **Current age** | | | | |
| 15–24 | 5,074(40.0) | 2,325(45.8) | Ref. | |
| 25–34 | 4,302(34) | 2,102(48.9) | 1.09(0.94, 1.27) | 0.231 |
| 35–49 | 3,296(26.0) | 1,780(54.0) | 1.41(1.21, 1.64) | <0.001 |
| **Residence** | | | | |
| Urban | 3,195(25.2) | 693(21.7) | Ref. | |
| Rural | 9,477(74.8) | 5,514(58.2) | 6.09(5.06, 7.32) | <0.001 |
| **Regions** | | | | |
| Major central regions | 11,251(87.8) | 5,760(51.2) | 4.35(3.44, 5.49) | <0.001 |
| Peripheral | 436(3.4) | 272(62.3) | 7.56(5.74, 9.97) | <0.001 |
| Metropolitans | 985(7.8) | 176(17.9) | Ref. | |
| **Religion** | | | | |
| Orthodox | 5,972(47.1) | 2,423(40.6) | Ref. | |
| Muslim | 3,504(27.7) | 2,107(60.1) | 1.97(1.63, 2.37) | <0.001 |
| Protestant | 2,951(23.3) | 1,523(51.6) | 1.29(1.04, 1.61) | 0.023 |
| Catholic | 92(0.7) | 57(61.9) | 1.82(0.79, 4.21) | 0.159 |
| Others | 153(1.2)) | 98(63.5) | 2.31 (1.21, 4.40) | 0.011 |
| **Marital status** | | | | |
| Unmarried | 3,409(26.9) | 1,380(40.5) | 0.79(0.61, 1.02) | 0.075 |
| Cohabited | 8,109(64.0) | 4,314(53.2) | 0.99(0.79, 1.24) | 0. 968 |
| Others | 1,154(9.1) | 513(44.5) | Ref. | |
| **Educational status** | | | | |
| No education | 5,394(42.6) | 3,472(64.3) | 4.94(4.05, 6.03) | <0.001 |
| Primary | 4,724(37.3) | 2,246(47.5) | 2.74(2.29, 3.28) | 0.011 |
| Secondary and higher | 2,555(20.1) | 489(19.2) | Ref. | |
| **Community level women illiteracy** | | | | |
| Low | 5,063(40.0) | 1,582(31.2) | Ref. | |
| High | 7,609(60.0) | 4,626(60.8) | 4.59 (3.72, 5.48) | <0.001 |
| **Occupational status** | | | | |
| Employed | 6,615(52.2) | 3,169(52.3) | Ref. | |
| Unemployed | 6,057(47.8) | 3,038(45.9) | 1.10(0.96, 1.27) | 0.313 |
| **Wealth index combined** | | | | |
| Poorest | 1,760(13.9) | 1,229(66.8) | 5.17(3.98, 6.71) | <0.001 |
| Poorer | 2,105(16.6) | 1,279(60.8) | 3.24(2.53, 4.16) | <0.001 |
| Middle | 2,377 (18.8) | 1,419(59.7) | 3.11(2.44, 3.97) | <0.001 |
| Richer | 2,574(20.3) | 1,305(50.7) | 2.13(1.67, 2.71) | <0.001 |
| Richest | 3,854(30.4) | 976(25.3) | Ref. | |
| **Community poverty** | | | | |
| Low | 5,678() | 2,013(35.5) | Ref | |
| High | 6,995() | 4,195(60.0) | 3.94 (3.23, 4.82) | <0.001 |
| **Family size** | | | | |
| < = 5 | 6,875 (54.2) | 3,051(44.4) | Ref. | |
| >5 | 5,797(45.8) | 3,157(54.5) | 1.12(0.88, 1.28) | 0.219 |
| **Sex of head of Household** | | | | |
| Male | 9,604() | 4,947(51.5) | 1.04(0.89, 1.21) | 0.608 |
| Female | 3,068() | 1,260(41.1) | Ref. | |
| **Parity** | | | | |
| Nulliparous | 4,365(34.5) | 1,786(40.9) | Ref. | |

*(Continued)*

**Table 2.** (Continued)

| Variable categories | Total Weighted frequency (%) [N = 12,672] | Inadequate HIV/AIDS-KAB (%) | COR[95% CI] | p-value |
|---|---|---|---|---|
| Primiparous | 1,669(13.2) | 758(45.4) | 1.17(0.97, 1.40) | 0.079 |
| Multiparous | 3,909(30.8) | 1,971(50.4) | 1.24(1.07, 1.43) | 0.004 |
| Grand multiparous | 2,730(21.5) | 1,693(62.0) | 1.61(1.32, 1.95) | <0.001 |
| **Contraceptive usage** | | | | |
| Yes | 3,445(27.2) | 1,495(43.4) | Ref. | |
| No | 9,227(72.8) | 4,713(51.1) | 1.19(1.06, 1.35) | 0.123 |
| **Autonomy in decision-making** | | | | |
| Low | 1,243 (9.8) | 881(70.8) | 2.03(1.62, 2.55) | 0.002 |
| Middle | 1,042(8.2) | 587(56.3) | 1.34(1.08, 1.66) | 0.04 |
| High | 10,387(82.0) | 4,740(45.6) | Ref. | |
| **Ease of distance to seek medical care** | | | | |
| Big problem | 5,973(47.1) | 3,614(60.5) | 1.39 (1.18, 1.64) | <0.001 |
| Not a big problem | 6,699(52.9) | 2,593(38.7) | Ref. | |
| **Ease of money to seek medical care** | | | | |
| Big problem | 6,527(51.5) | 3,791(58.1) | 1.53(1.32, 1.78) | <0.001 |
| Not a big problem | 6,145(48.5) | 2,416(39.3) | Ref. | |
| **Listen to radio** | | | | |
| Not at all | 8,049(63.5) | 4,555(56.6) | 2.35(1.94, 2.84) | <0.001 |
| Less than once a week | 2,317(18.3) | 941(40.6) | 1.51(1.21, 1.87) | <0.001 |
| At least once a week | 2,305(18.2) | 712(30.9) | Ref. | |
| **Watching TV** | | | | |
| Not at all | 8,627 (68.1) | 5,113(59.3) | 3.96(3.19, 4.91) | <0.001 |
| Less than once a week | 1,690(13.3) | 651(38.5) | 2.11(1.69, 2.63) | <0.001 |
| At least once a week | 2,355(18.6) | 443(18.8) | Ref. | |
| **Reading newspaper** | | | | |
| Not at all | 10,663(84.2) | 5,776(54.2) | 3.09(2.10, 4.57) | <0.001 |
| Less than once a week | 1,438(11.3) | 298(20.7) | 1.04(0.69, 1.55) | 0.844 |
| At least once a week | 571(4.5) | 134(23.4) | Ref. | |
| **Community media exposure** | | | | |
| Low | 7,760(61.2) | 4,648(59.9) | 4.37 (3.59, 5.31) | <0.001 |
| High | 4,913(38.8) | 1,560(31.8) | Ref. | |
| **Own mobile phone** | | | | |
| No | 8,753(69.1) | 5,193(59.3) | 2.93(2.50, 3.52) | <0.001 |
| Yes | 3,919(30.9) | 1,015(25.9) | Ref. | |
| **Covered by Health Insurance** | | | | |
| Yes | 716(5.7) | 304(42.5) | Ref. | |
| No | 11,956(94.3) | 5,903(49.4) | 1.11(0.85, 1.45) | 0.443 |
| **Visit health facility within the last 1 year** | | | | |
| Yes | 5,634(55.5) | 2,342(41.6) | Ref. | |
| No | 7,036(44.5) | 3,866(54.9) | 1.52 (1.33, 1.74) | <0.001 |

Ref.: Reference, COR: Crude Odds Ratio

**Hot spot (Getis-Ord Gi\*) analysis.** As per hot spot analysis, Somali, and most parts of Afar regions were identified as hot spots for women with inadequate HIV/AIDS knowledge. The central part of Addis Ababa, as well as several parts of the Oromia region, on the other hand, were cold spots (Fig 5).

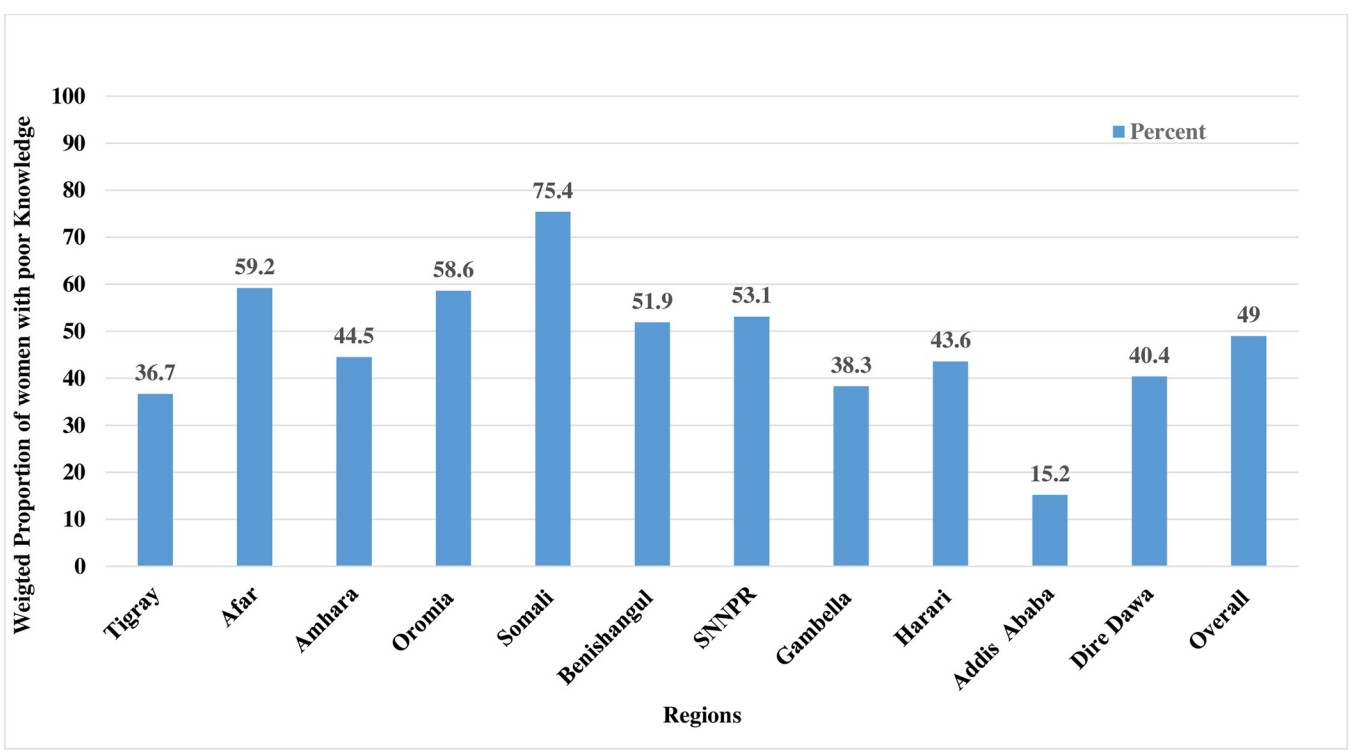

**Fig 2. Regional disparity in the proportion of women with inadequate HIV/AIDS-KAB, EDHS 2016.**

**Spatial interpolation.** The spatial distribution of inadequate HIV/AIDS-KAB for locations where data were not obtained has been estimated using the standard Kriging technique. Accordingly, the highest estimated prevalence of inadequate KAB (red shaded) was observed in southern Somali, North and South Afar, and parts of Dire Dawa (Fig 6).

**Spatial scan statistical (SaTScan) analysis.** The SaTScan geographical analysis showed ten statistically significant groups of SaTScan clusters with a high proportion of women with inadequate KAB; the prevalence of inadequate HIV/AIDS-KAB was higher inside the SaTScan circular window than outside one. Furthermore, the analysis came across 306 significant clusters that accounted for inadequate knowledge. The first most likely cluster was located at

**Table 3. The level of HIV/AIDS among Ethiopia women, EDHS 2016.**

| Items used to assess overall KAB | Yes (%) | No (%) |
|---|---|---|
| The use of condoms to reduce HIV transmission | 8,329(65.7) | 4,343(34.7) |
| The use of Abstinence to reduce HIV transmission | 9,701(76.5) | 2,971(23.45) |
| Getting HIV by mosquito bite | 5,830(46.0) | 6,843(54.0) |
| People can get HIV by sharing food with a person who has AIDS | 2,414(19.0) | 10,258(81.0) |
| A healthy-looking person can have and transmit HIV | 8,619(68.0) | 4,054(32.0) |
| HIV transmitted during pregnancy | 10,208(80.6) | 2,464(19.4) |
| HIV transmitted during delivery | 11,001(86.8) | 1,671(13.2) |
| HIV transmitted during breastfeeding | 11,619(91.7) | 1,054(8.3) |
| Know the place to get an HIV test | 9,776(77.2) | 2,896(22.8) |
| Can get HIV by witchcraft or supernatural means | 2,415(19.0) | 10,258(81.0) |
| Drugs to avoid HIV transmission to the baby during pregnancy | 8,031(63.4) | 4,642(36.6) |

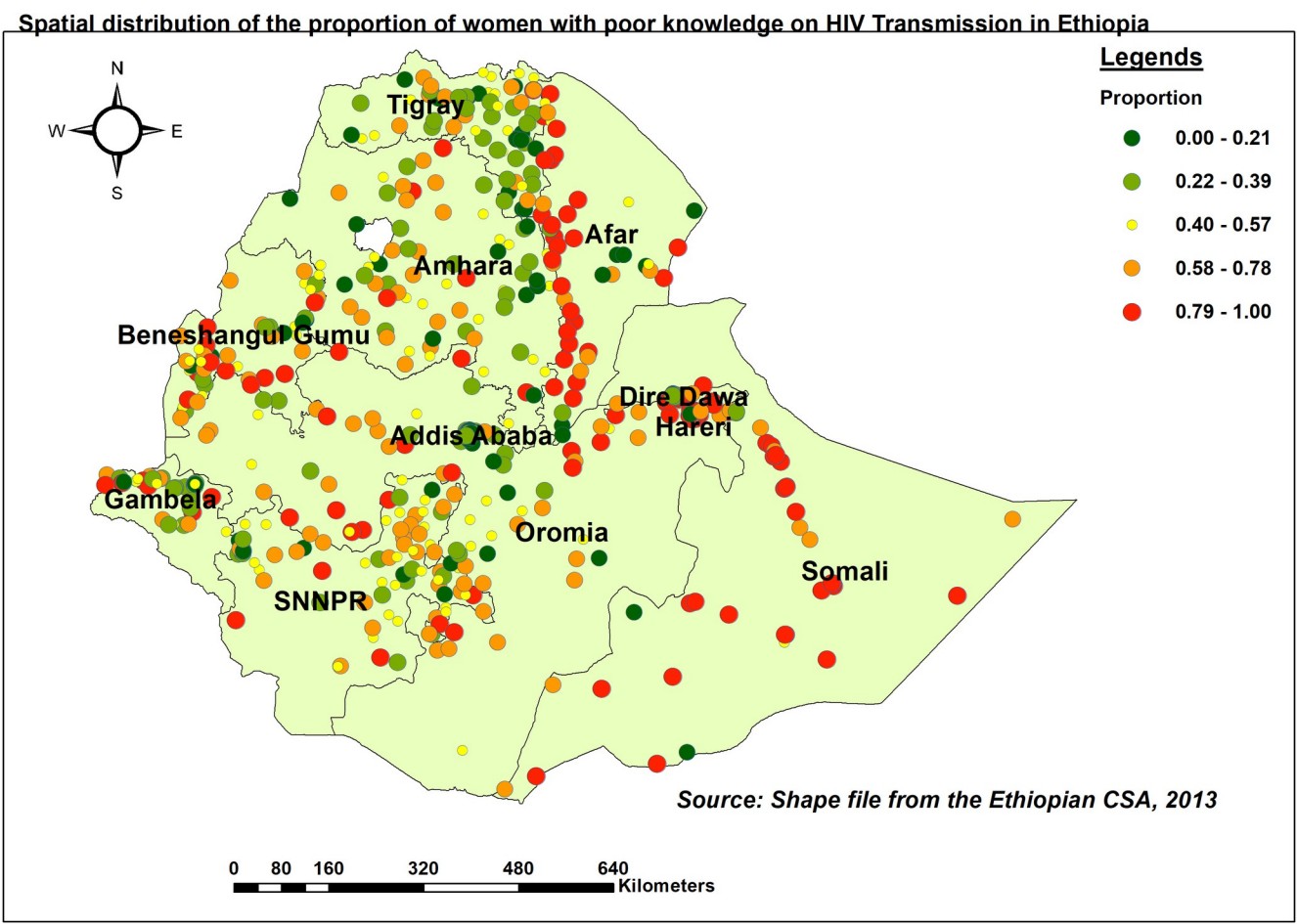

**Fig 3. Spatial distribution of women with inadequate HIV/AIDS-KAB, EDHS 2016.**

geographical coordinates (7.717178 N, 46.991580 E) with a 560.67 Km radius, LLR of 96.86, and Relative risk (RR = 1.74); women in the area were 74% more likely to have inadequate KAB than their counterparts outside the area (**S2 File**).

## Results of spatial regression

**The global ordinary least square (OLS) analysis results.** The OLS model is the initial step towards selecting relevant predictors for the geographical variability of inadequate HIV/AIDS-KAB. Accordingly, inadequate KAB was found to be associated with a lack of formal education, living in the poorest wealth quintile, a lack of autonomy in decision-making, and never listening to the radio and watching television. There was no evidence of multicollinearity among the predictors (minimum and maximum VIF of 1.14 and 2.05, respectively). Furthermore, the null model explained 63.1% of the variation in inadequate knowledge (adjusted $R^2$ = 0.631). Jarque-Bera statistics with p-values greater than 0.05 (p = 0.118) indicate that the model prediction was not skewed. The Koenker statistics in the model, on the other hand, revealed a statistically significant p-value (p<0.001), indicating that the regression model is inconsistent across the study area, suggesting that the local (GWR) model is required to appropriately estimate the model parameters (Table 4).

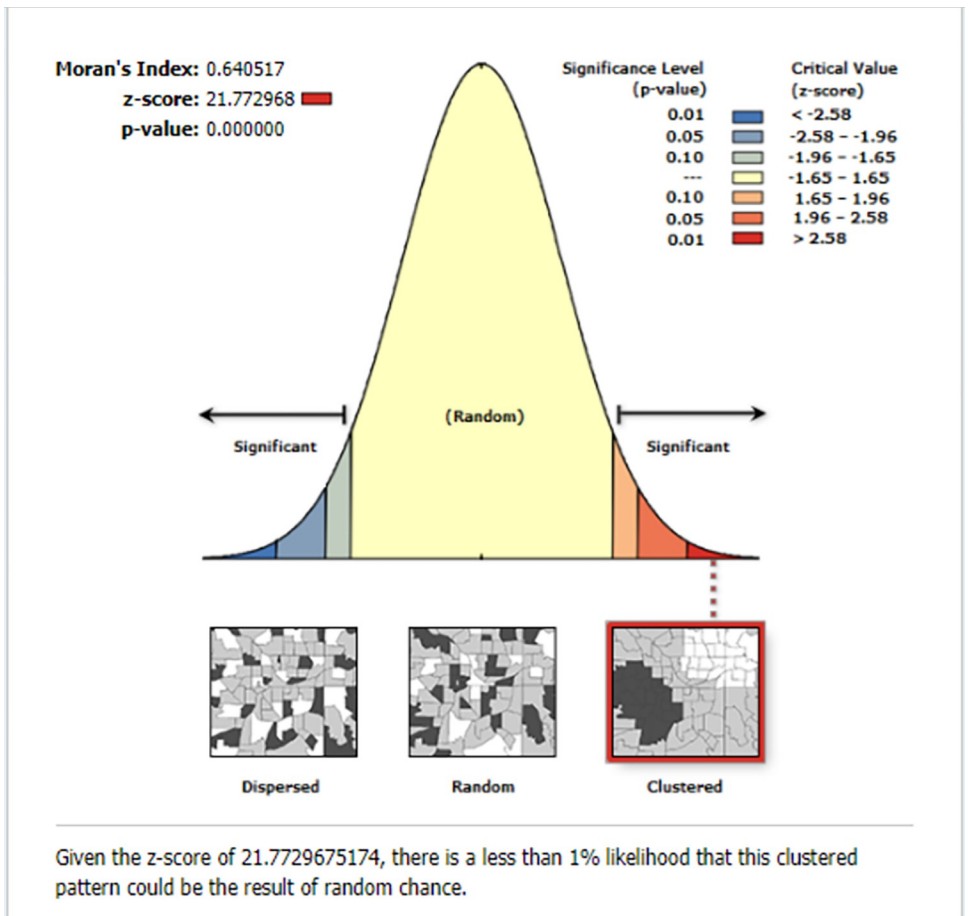

**Fig 4. The global spatial autocorrelation of inadequate HIV/AIDS-KAB, EDHS 2016.**

**Geographically weighted regression (GWR) analysis.** The GWR analysis outperformed the global model (OLS) significantly. The AICc value of -443.03 in the OLS model dropped to -540.27 in the GWR, while the adjusted R2 value of 0.631 in the OLS rose to 0.693 in the GW; demonstrating the necessity of moving from global (OLS) to local (GWR) regression to identify spatial predictors of inadequate KAB (Tables 4 and 5).

The GWR model was highly explained (with the highest range of R-square) in Tigray, central Afar, northern parts of Somali, central parts of Dire Dawa and Harari, and eastern borders of Amhara and Addis Ababa (Fig 7).

In most parts of Somali, Benishangul Gumuz, and Gambella regions, and the eastern parts of SNNPR, as the proportion of women without formal education rose, so did the proportion of women with inadequate KAB (adaptive bandwidth = 159212.73, R2 adjusted = 0.588). On the other hand, in most portions of the Amhara, Tigray, and Afar regions, the association between lack of formal education and inadequate comprehensive HIV/AIDS-KAB was found to be relatively weaker (Fig 8).

This study also highlights the space-dependent relationship between inadequate HIV/AIDS knowledge and the wealth index (adaptive bandwidth = 159212.73, R2 adjusted = 0.448383). As the proportion of women living in the poorest wealth quintile increases, the likelihood of having inadequate knowledge also increases in Somali, Benishangul Gumuz, and vast areas of SNNPR (Fig 9).

**Hot Spot and Cold Spots of inadequate comprehensive HIV/AIDS knowledge among Ethiopian Women**

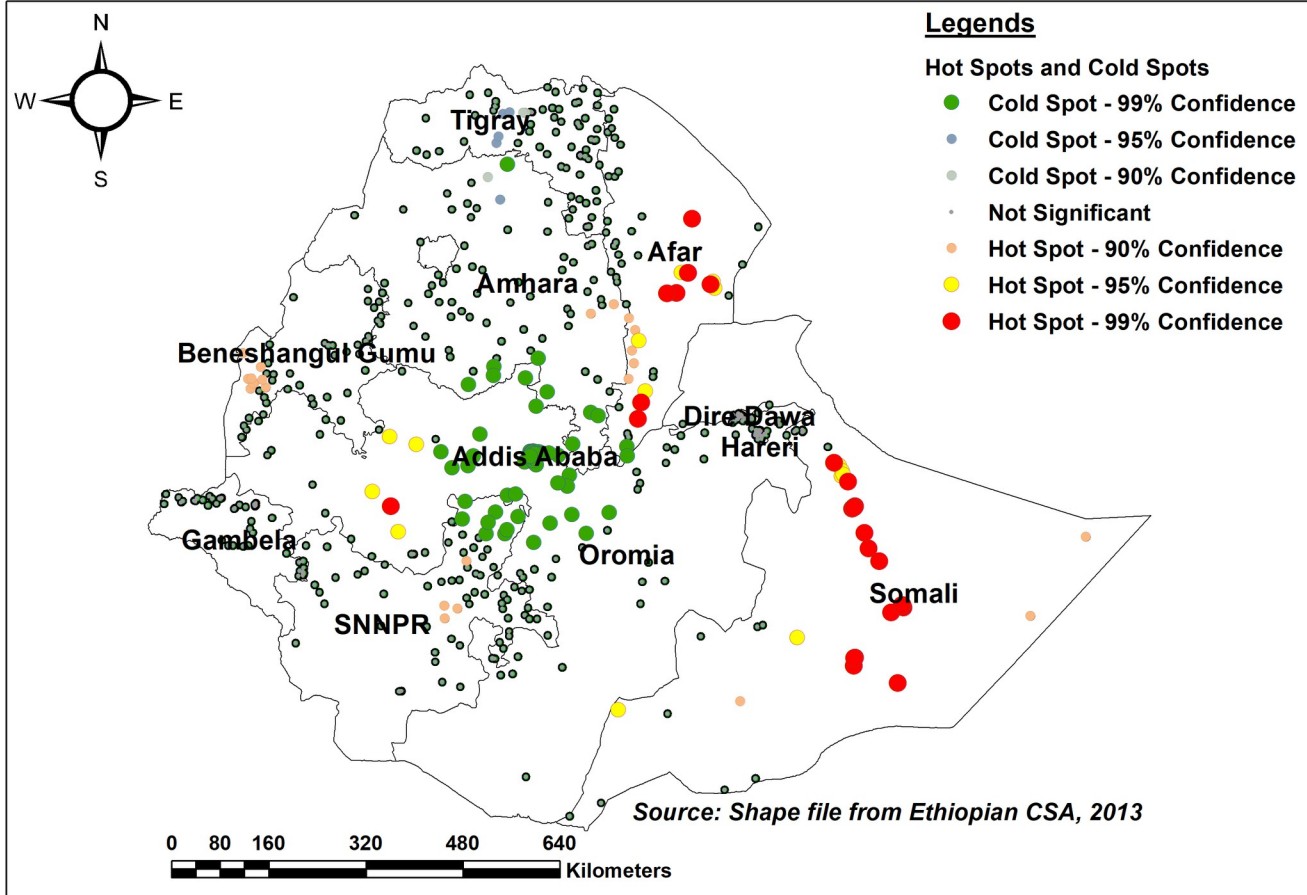

**Fig 5. Hot spot and cold spot of inadequate HIV/AIDS knowledge, EDHS 2016.**

Lack of autonomy in decision-making had a strong link with inadequate comprehensive HIV/AIDS-KAB. As the proportion of non-autonomous women rose, so did the level of inadequate KAB on HIV in Somali followed by Afar, Benishangul Gumuz, and north of the SNNPR (adaptive bandwidth = 159212.73, R2 adjusted = 0.279) (Fig 10).

As the proportion of women who never listened to the radio increased, the proportion of inadequate HIV/AIDS-KAB in various parts of SNNPR, southwest Somali, the southern part of Benishangul Gumuz, and northern parts of Gambella (adaptive bandwidth = 215518.028, R2 adjusted = 0.402845). The red-colored clustered points indicate areas where the coefficients were largest, which in turn indicates the strong positive association between the two variables (Fig 11).

Similarly, as the proportion of women who never watched TV increased, the proportion of inadequate HIV/AIDS-KAB in various parts of Somali, Dire Dawa, Harari, the western borders of Oromia, and the central part of Benishangul Gumuz (adaptive bandwidth = 184107.541, R2 adjusted = 0.583) (Fig 12). A negative coefficient in GWR suggests that there is a negative relationship between the predictor variable and the response variable at that specific location.

As the number of women who did not visit health facilities in the previous 12 months increased, so did the proportion of people with low HIV/AIDS-KAB in most areas of Somali region (adaptive bandwidth = 159212.73, R2 adjusted = 0.254) (Fig 13).

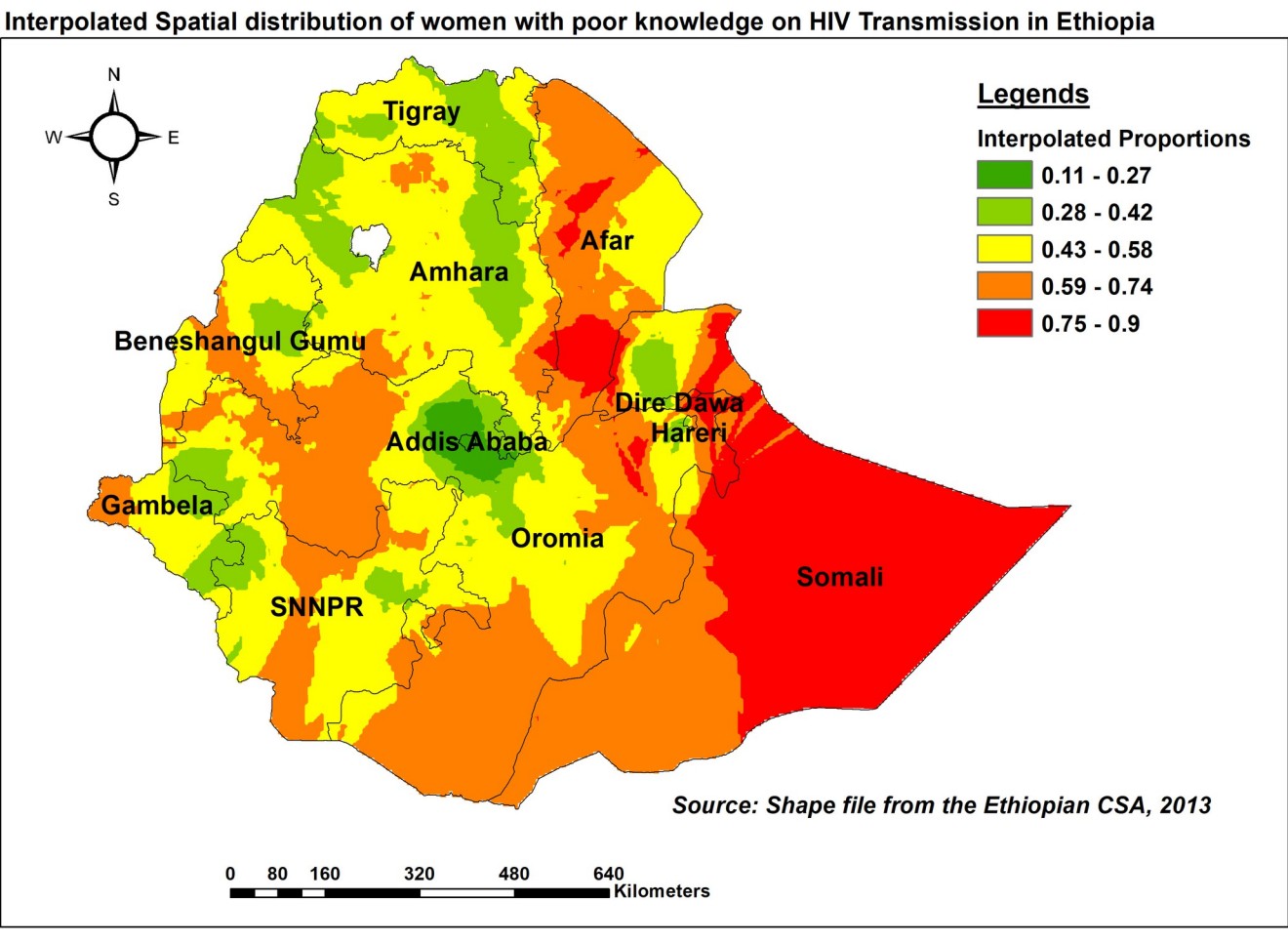

**Fig 6. Ordinary Kriging interpolation of the spatial distribution of inadequate HIV/AIDS-KAB knowledge, EDHS 2016.**

### Results of multilevel mixed-effect logistic regression

**Random effect (measures of variation).** The ICC in the null model revealed that 33.1% of the variability in inadequate HIV/AIDS-KAB was explained by between-cluster variability (ICC = 0.331, p<0.001). The median odds ratio (MOR) indicates the amount to which the risk of having inadequate HIV/AIDS-KAB was determined by clusters and is hence suitable for assessing contextual factors. In the current study, a MOR of 3.34 in the null model revealed that if we randomly selected a woman from two different clusters, the woman from the cluster with a higher rate of inadequate HIV/AIDS-KAB had 3.34 times higher odds of having poor KAB as compared to the woman from the cluster with a higher KAB. Furthermore, a proportionate change in variance (PCV) of 81.25% in the full model suggests that individual and community-level factors together accounted for 81.25% of the variability seen in the null model. As we moved from model 1 (the empty model) to model 4 (the full model), the values of AIC, BIC, and Deviance declined, indicating that the final model fitted throughout the study had appropriate goodness of fit. Finally, the fourth model with the lowest deviance (13307.2) was selected as the best model fit for the study (Table 6).

**Fixed effects: Predictors of inadequate knowledge about HIV transmission.** In the multilevel multivariable logistic regression analysis, wealth index, educational status, autonomy in decision-making, media exposure, mobile ownership, visit to health facilities, residence, and

**Table 4. Summary of OLS results for inadequate HIV/AIDS-KAB, EDHS 2016.**

| Variable | Coefficient | SE | t-Statistic | Probability | Robust SE | Robust t-statistics | Robust probability | VIF |
|---|---|---|---|---|---|---|---|---|
| Intercept | 1.013 | 0.024 | 42.41 | 0.001 | 0.023 | 43.30 | <0.001 | — |
| Women without formal education | -0.277 | 0.036 | -7.55 | <0.001 | 0.043 | -6.44 | <0.001 | 2.05 |
| Women living in the poorest wealth quintile | -0.083 | 0.027 | -3.06 | 0.002 | 0.030 | -2.72 | 0.006 | 1.80 |
| Non-autonomous women | -0.211 | 0.053 | -3.94 | <0.001 | 0.058 | -3.62 | 0.0003 | 1.28 |
| Women who never listen to radio | -0.109 | 0.038 | -2.84 | 0.005 | 0.041 | -2.63 | 0.009 | 1.75 |
| Women who never watch television | -0.153 | 0.033 | -4.64 | <0.001 | 0.034 | -4.40 | <0.001 | 1.72 |
| Women who didn't visit a health facility within the last 12 months | -0.249 | 0.036 | -6.81 | <0.001 | 0.047 | -5.28 | <0.001 | 1.14 |
| **OLS Diagnostics** | | | | | | | | |
| Number of Observations: | 613 | | Akaike's Information Criterion (AICc) | | | | -443.03 | |
| Multiple R-Squared | 0.636 | | Adjusted R-Squared [d]: | | | | 0.631 | |
| Joint F-Statistic | 131.89 | | Prob(>F), (7,599) degrees of freedom | | | | <0.001 | |
| Joint Wald Statistic | 2118.68 | | Prob(>chi-squared),(7) degrees of freedom | | | | <0.001 | |
| Koenker (BP) Statistic | 30.61 | | Prob(>chi-squared), (7) degrees of freedom | | | | <0.001 | |
| Jarque-Bera Statistic | 123.21 | | Prob(>chi-squared), (2) degrees of freedom | | | | 0.118 | |

region were identified as significant determinants of the level of comprehensive HIV/AIDS-KAB (Table 7).

Women from rural areas were 1.87 times more likely to have inadequate comprehensive HIV/AIDS-KAB (AOR = 1.87; 95% CI: 1.48, 3.85) than their urban counterparts. Similarly, the likelihood of having inadequate HIV/AIDS-KAB was 62% higher among rural women than among urban women (AOR = 1.62; 95% CI: 1.18, 2.22). Women with no formal education were 2.66 (AOR = 2.66; 95% CI: 2.04, 3.48) times more likely to have inadequate HIV/AIDS-KAB than women with secondary education or higher educational status. The odds of inadequate KAB were 1.56 (AOR = 1.56; 95% CI: (1.02, 2.39) and 1.50 (AOR = 1.50; 95% CI: 1.17, 1.92) higher among women who never listen to the radio and watched television, respectively. Non-autonomous women had 80% (AOR = 1.80; 95% CI: (1.43, 2.28) higher likelihood of having inadequate KAB as compared to autonomous women (Table 7).

## Discussion

This study assessed the prevalence, spatial variation, and determinants of inadequate HIV/AIDS-KAB among Ethiopian women. Nearly half (48.9%), of women had inadequate HIV/AIDS-KAB. There was significant spatial variation in the prevalence of comprehensive knowledge across small geographical areas. The likelihood of having inadequate HIV/AIDS-KAB was higher among women living in the poorest wealth quintile, residing in rural areas, with no

**Table 5. Summary of GWR model fitted with potential spatial predictors of inadequate HIV/AIDS-KAB, EDHS 2016.**

| List of variables | Women without formal education, living in the poorest wealth quintile, non-autonomous in decision making, who never listen to radio, who never watch television, and who didn't visit a health facility within the last 12 months |
|---|---|
| Residual squares | 13.044 |
| Effective number | 49.567 |
| Sigma | 0.152 |
| AICc | -540.27 |
| Multiple R-Squared | 0.718 |
| Adjusted R-Squared | 0.693 |

GWR analysis for spatial predictors of having inadequate knowledge on HIV transmission among women in Ethiopia

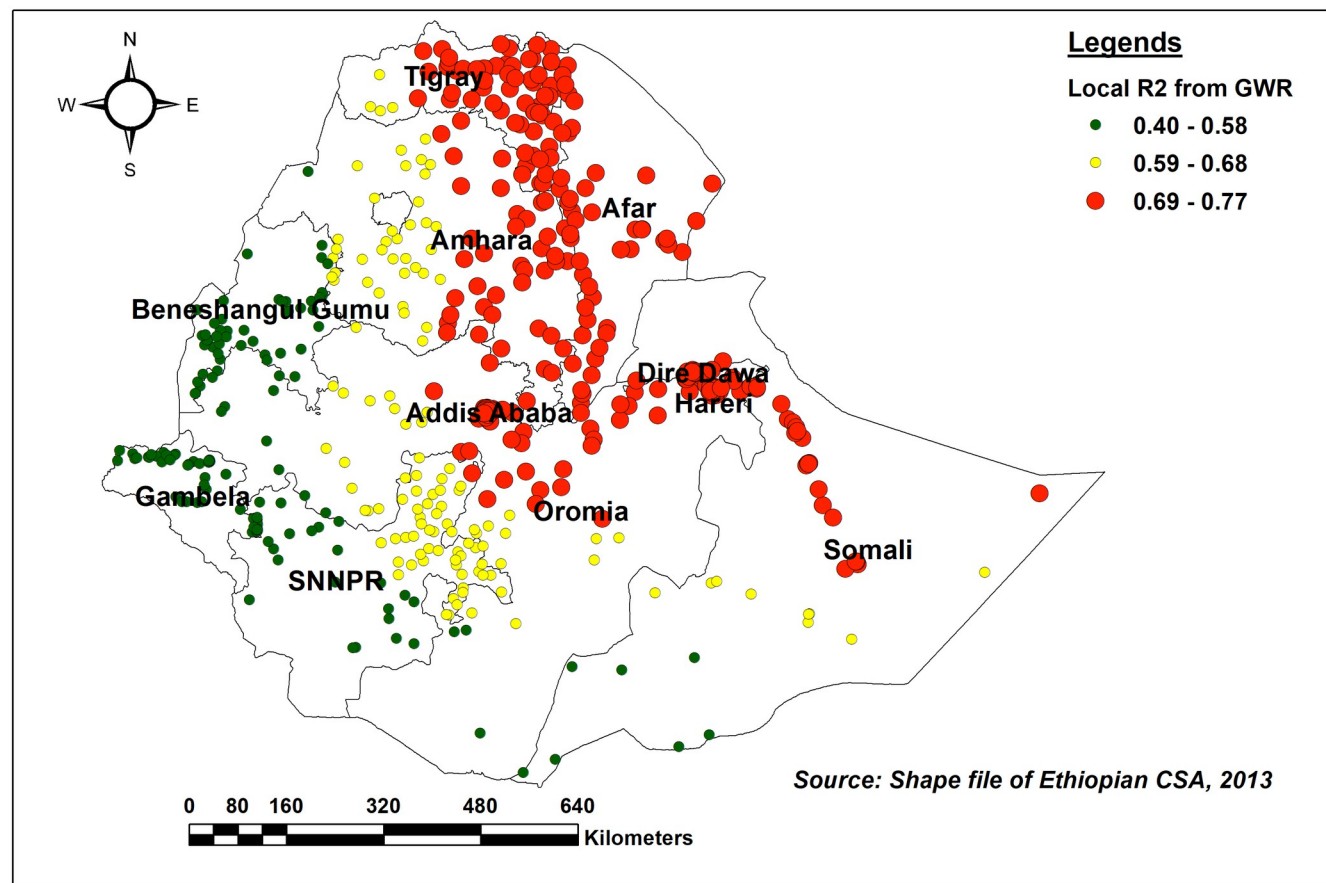

**Fig 7. The spatial mapping of local adjusted R-square of the GWR analysis for selected spatial predictors of inadequate HIV/AIDS-KAB, EDHS 2016.**

formal education, who were non-autonomous in decision-making, not exposed to media (radio and television), who didn't have mobile and didn't visit health facilities.

As this finding was based on a composite index of the three domains (knowledge, attitude, and behaviour), we encountered a dearth of studies conducted comparably, therefore we tried to discuss using studies that focused on the knowledge and attitude (misconceptions) domain. Accordingly, the level of inadequate comprehensive HIV/AIDS-KAB in this study was consistent with a study in Uganda (48.1%) [45], but higher than studies conducted in Indonesia (46.4%) [46], China (19.1%) [47], Tajikistan (12.95%) [48], and Pakistan (32%) [49] and lower than findings in Kenya (53.7%) [50], India (74.2%) [51], Iraq (91.2%) [52], Malawi (57.8%) [53] or low and middle-income countries (70.7%) [14] and Sub-Shahran Africa(61.44%) [11]. Disparities in socioeconomic status, educational opportunities and attainment, cultural and societal attitudes towards sex education and HIV/AIDS discussions, gender disparities, access to healthcare facilities and the quality of health services, and the availability, accessibility, and dissemination of accurate HIV/AIDS information could all contribute to the disparities [14, 54, 55]. Thus, the Ethiopian government and stakeholders in the health sector need to work jointly to reduce the high prevalence of inadequate comprehensive HIV/AIDS knowledge through a multifaceted approach: public health campaigns, education programmes, and policy initiatives that can play a significant role in promoting accurate information and awareness.

## Proportion of women without formal education

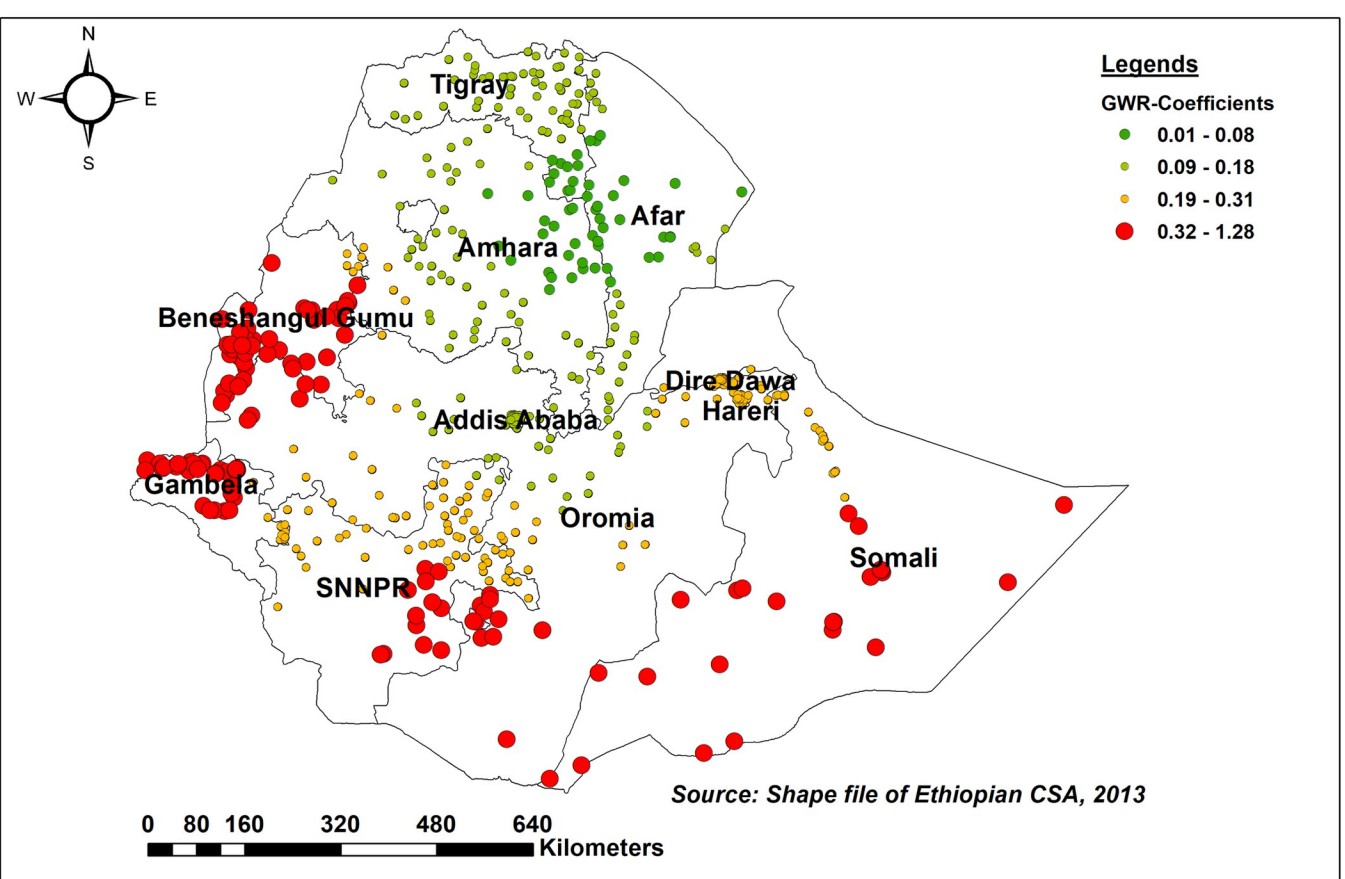

**Fig 8. GWR coefficients of the proportion of women lacking formal education for predicting inadequate HIV/AIDS-KAB, EDHS 2016.**

The study pointed to the need for special emphasis during resource allocation and targeted interventions in Somali and the southern parts of Afar, which showed relatively higher inadequate comprehensive HIV/AIDS-KAB (significant spatial clusters [Hot Spots]). The interventions could be strengthening tailored educational campaigns, and community engagement initiatives including support groups, media infrastructure, and coverage in the local language. Incorporating culturally sensitive educational materials and disseminating them through local community leaders and influencers and mobile health initiatives, including SMS campaigns can be employed to ensure widespread access to accurate information. Strengthening healthcare infrastructure in these regions is essential, enhancing accessibility and providing training for healthcare professionals. Higher inadequate comprehensive HIV/AIDS-KAB in Somali and the southern parts of Afar could be attributed to cultural factors, limited access to healthcare facilities and services, limited possibilities for women's empowerment, and the way of life (pastoral communities), impeding the dissemination of health information. Ongoing research and monitoring efforts should be conducted to understand unique drivers for low knowledge and solutions to address them.

Having no formal education was identified as both an individual- and spatial predictor of inadequate comprehensive HIV/AIDS-KAB, which is supported by previous studies [11, 19, 45, 49, 56]. This could be because women who have no formal education may have limited

**GWR analysis for spatial association between living in the poorest wealth quintile and inadequate knowledge**

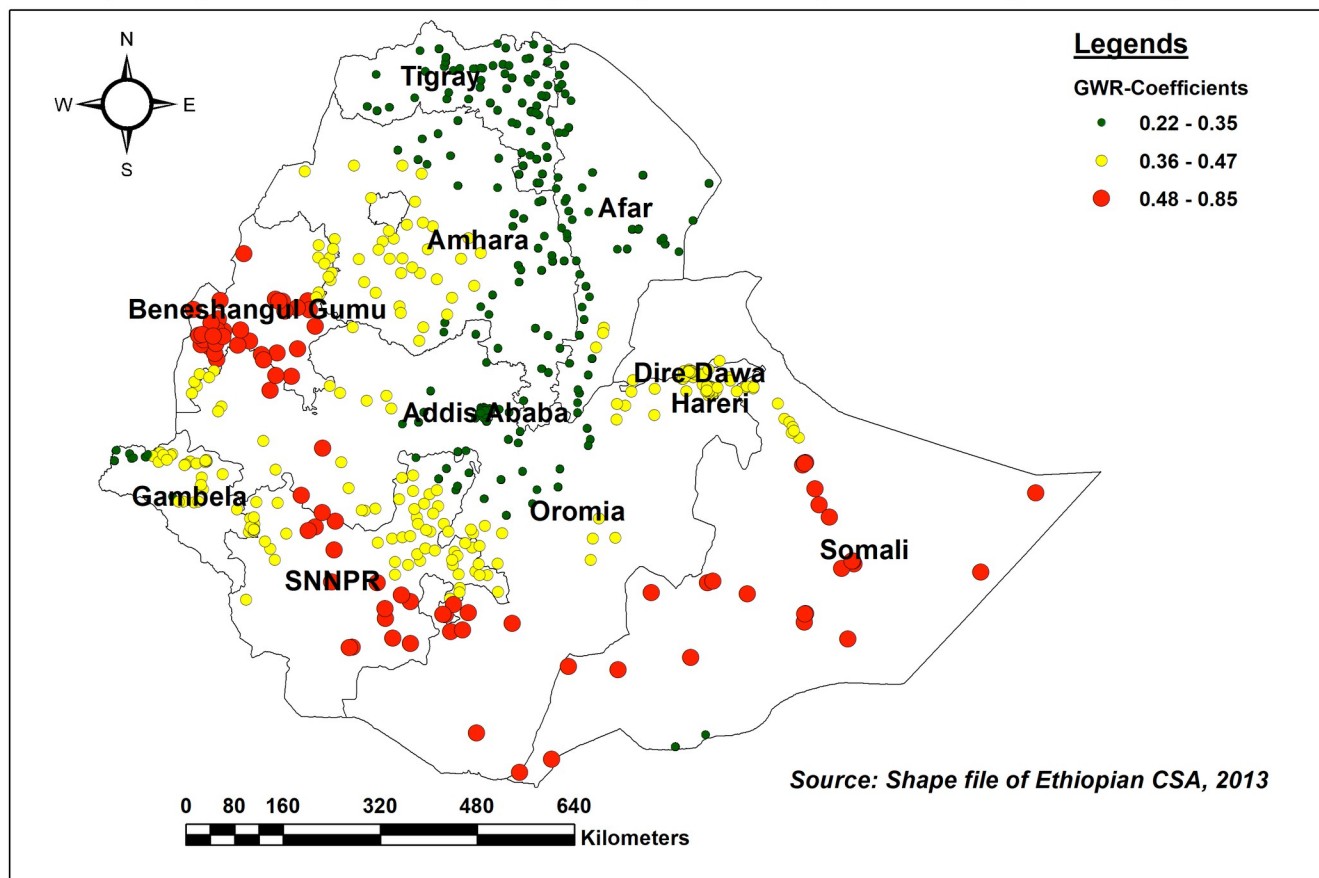

**Fig 9. GWR coefficients of the proportion of women living in the poorest wealth quintile for predicting inadequate HIV/AIDS-KAB, EDHS 2016.**

access to HIV/AIDS information, hampering their ability to obtain correct and up-to-date information. In addition, they may lack empowerment, hindering their ability to seek information autonomously, hesitant to seek information or services due to fear of judgment and discrimination, and restricted mobility might lead to a lack of access to essential health information [49, 57]. Thus, a concerted effort is required to enhance access to education and emphasise the provision of HIV/AIDS education tailored to this population group.

In tandem with studies conducted in SSA countries [11], Pakistan [49], Malawi [53], and Uganda [45] inadequate HIV/AIDS-KAB was higher among women living in the poorest wealth quintile. The possible justifications could be women in the poorest wealth quintile may have limited access to media and healthcare services, including HIV testing and counselling [58]. In addition, the financial challenges can lead to survival-focused priorities, leaving little room for individuals to prioritize health education and could result in inadequate knowledge. Thus, tailored interventions that consider the specific challenges faced by women in the poorest wealth quintile can contribute to improving comprehensive knowledge and reducing the spread of the virus in these communities.

Women's autonomy in decision-making in accessing information and healthcare has significant implications, particularly for knowledge and attitude about HIV/AIDS, a disease that disproportionately affects women. In our study, lacking autonomy in decision-making was

**GWR analysis for spatial association between being  non-autonomous and inadequate knowledge of HIV transmission**

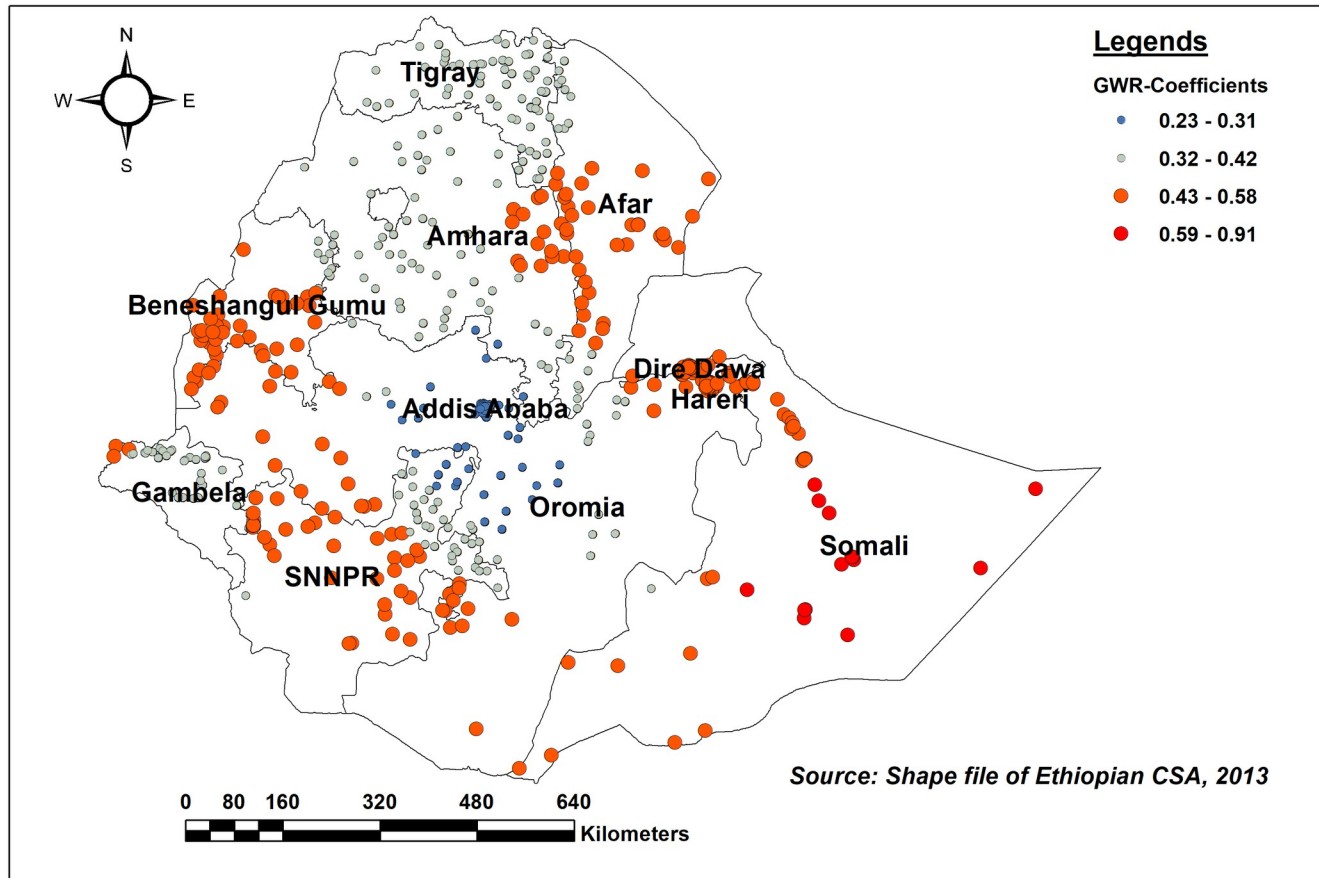

**Fig 10.  GWR coefficients of the proportion of non-autonomous women for predicting the level of inadequate HIV/AIDS-KAB, EDHS 2016.**

found to increase the odds of inadequate HIV/AIDS-KAB. It was also identified as a significant spatial determinant. Studies conducted elsewhere [19, 49] supported the finding. This could be due to the lack of autonomy could result in economic dependence on male partners or family members, distorted power dynamics, and limited access to health information, which may deter women from seeking HIV/AIDS information. Thus, increasing gender equality, expanding access to education and healthcare, confronting cultural stigmas, and empowering women to make informed health decisions should be the necessary steps by the government to increase women's autonomy.

In line with studies conducted in SSA countries [11], India [51], and Pakistan [49], lack of media exposure (Radio, Television, and mobile phone) was associated with inadequate comprehensive HIV/AIDS knowledge. This might be due to women who lack media exposure might face limited access to health information and updates regarding modes of transmission and prevention which can hinder the acquisition of comprehensive knowledge about HIV/AIDS. In addition, mobile phones allow women to stay connected and communicate with family, friends, and support networks both in voice conversation and social media platforms which can provide them with a variety of information regarding HIV/AIDS.

Finally, the likelihood of inadequate comprehensive HIV/AIDS-KAB was higher among rural women, which was supported by similar studies [11, 45, 49, 56]. This might be due to

**GWR analysis for spatial association between never listening to radio and inadequate knowledge**

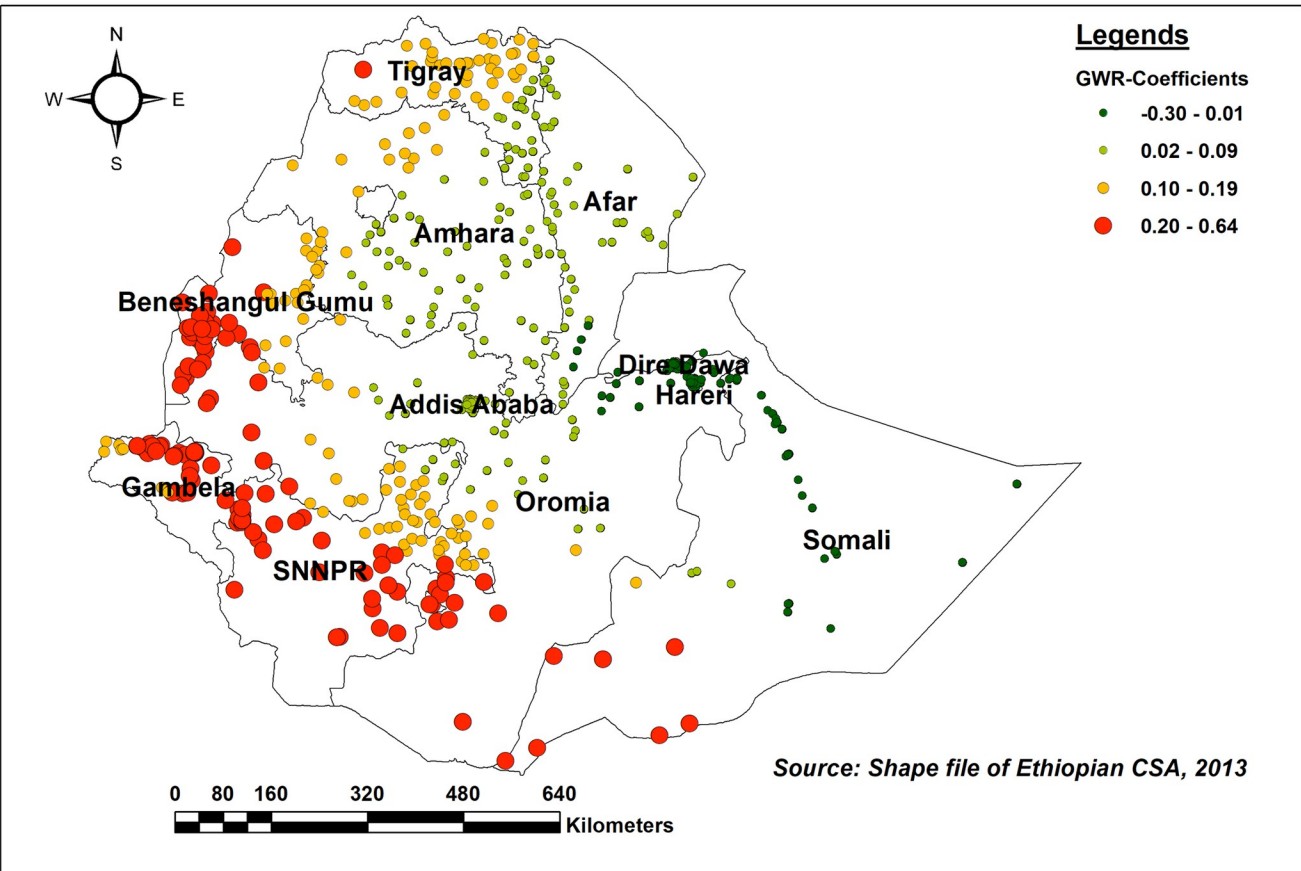

**Fig 11. GWR coefficients of the proportion of women who never listen to the radio for predicting the level of inadequate HIV/AIDS-KAB, EDHS 2016.**

rural women having economic challenges, and limited access to educational resources, healthcare facilities, and information campaigns about HIV/AIDS compared to urban areas, which can hinder the dissemination of accurate information about prevention, transmission, and misconceptions. In addition, they might have limited decision-making power or autonomy, which can impact their ability to access education and healthcare. As a result, efforts should be made to improve HIV/AIDS-KAB by emphasising culturally relevant education, widening access to healthcare facilities, empowering women via education and community participation, and tackling socioeconomic inequities.

This study had several strengths. First, the findings were based on an analysis of nationally representative data, thereby enhancing the generalizaAs the findings rely on both spatial and multilevel logistic regression analyses, they might help government and programme planners in developing geographic, individual, and community-focused public health interventions to improve women's comprehensive knowledge of HIV/AIDS. Furthermore, unlike previous studies that mostly relied on the knowledge domain, this study attempted to examine the composite of all three domains (knowledge, attitude, and behaviour), which may provide more comprehensive evidence to stakeholders. However, this finding should be considered in light of some limitations. Due to the cross-sectional nature of the data, it can be difficult to handle recall bias and figure out the causal relationship between the outcome of interest and

**GWR analysis for spatial association between never watching TV and inadequate knowledge**

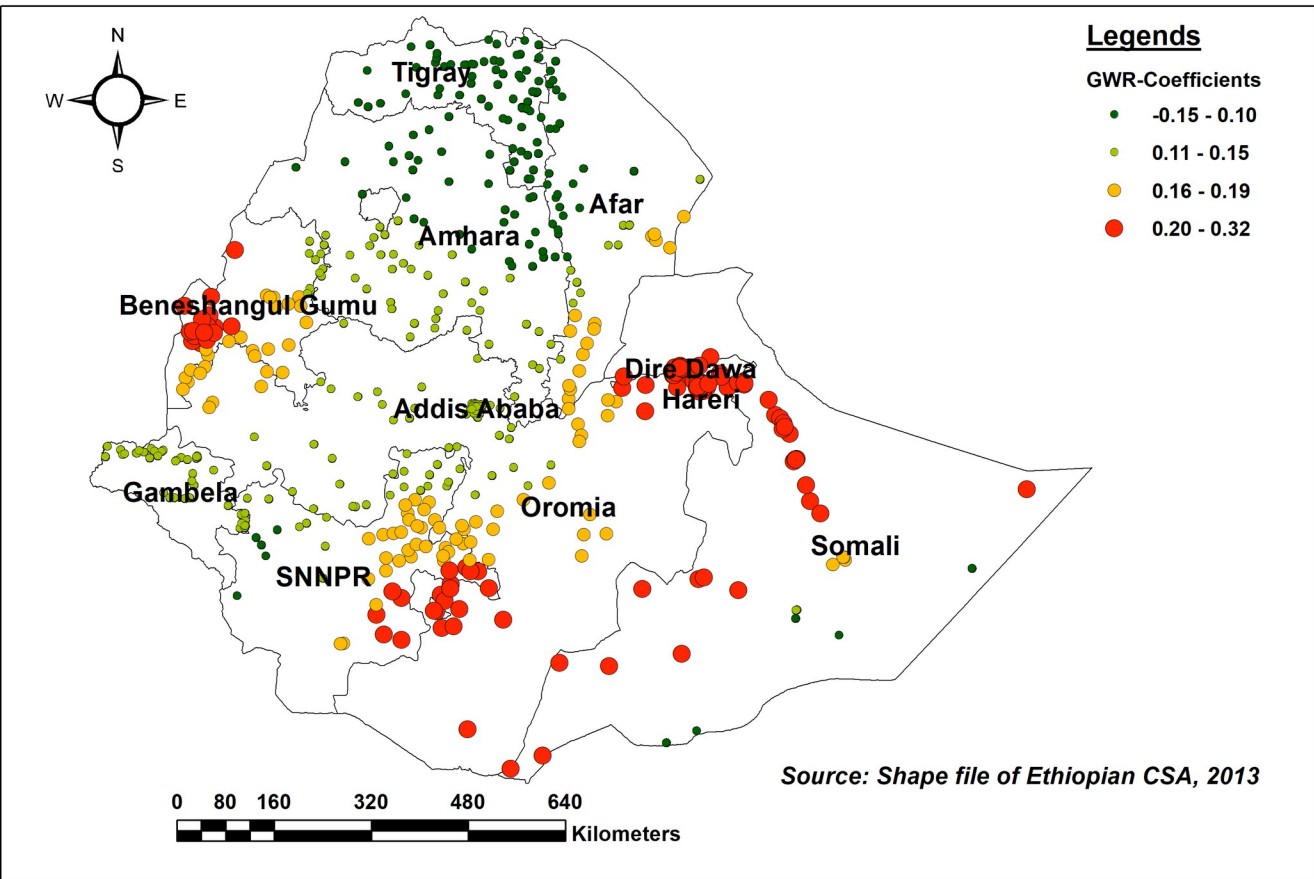

**Fig 12. GWR coefficients of the proportion of women who never watch TV for predicting the level of inadequate HIV/AIDS-KAB, EDHS 2016.**

covariates. Furthermore, given that the community-based surveys attempted to uncover inadequate HIV/AIDS comprehensive knowledge through a series of interviews, they may have delved into sensitive issues, and thus it is important to acknowledge the possibility of social desirability bias.

## Conclusion

About half of women in Ethiopia had inadequate HIV/AIDS-KAB, with significant spatial variation across regions identified. Higher inadequate HIV/AIDS-KAB was identified among women with the poorest wealth quintile, being a rural resident, having no formal education, lacking autonomy in decision-making, not being exposed to media (radio and television), not having a mobile phone, and not visiting health facilities. Thus, stakeholders in the healthcare sector need to work in the provision of HIV/AIDS education tailored to women who lack formal education, live in rural areas, and live in the poorest wealth quintile by as static and outreach service provision. Local and international collaborations should be strengthened to increase women's autonomy by promoting gender equality and empowering women to access HIV/AIDS health information. The media agencies also prioritise their accessibility to a broader portion of the population by providing health information tailored to women of

**GWR analysis for spatial association between not visiting health facility and inadequate knowledge**

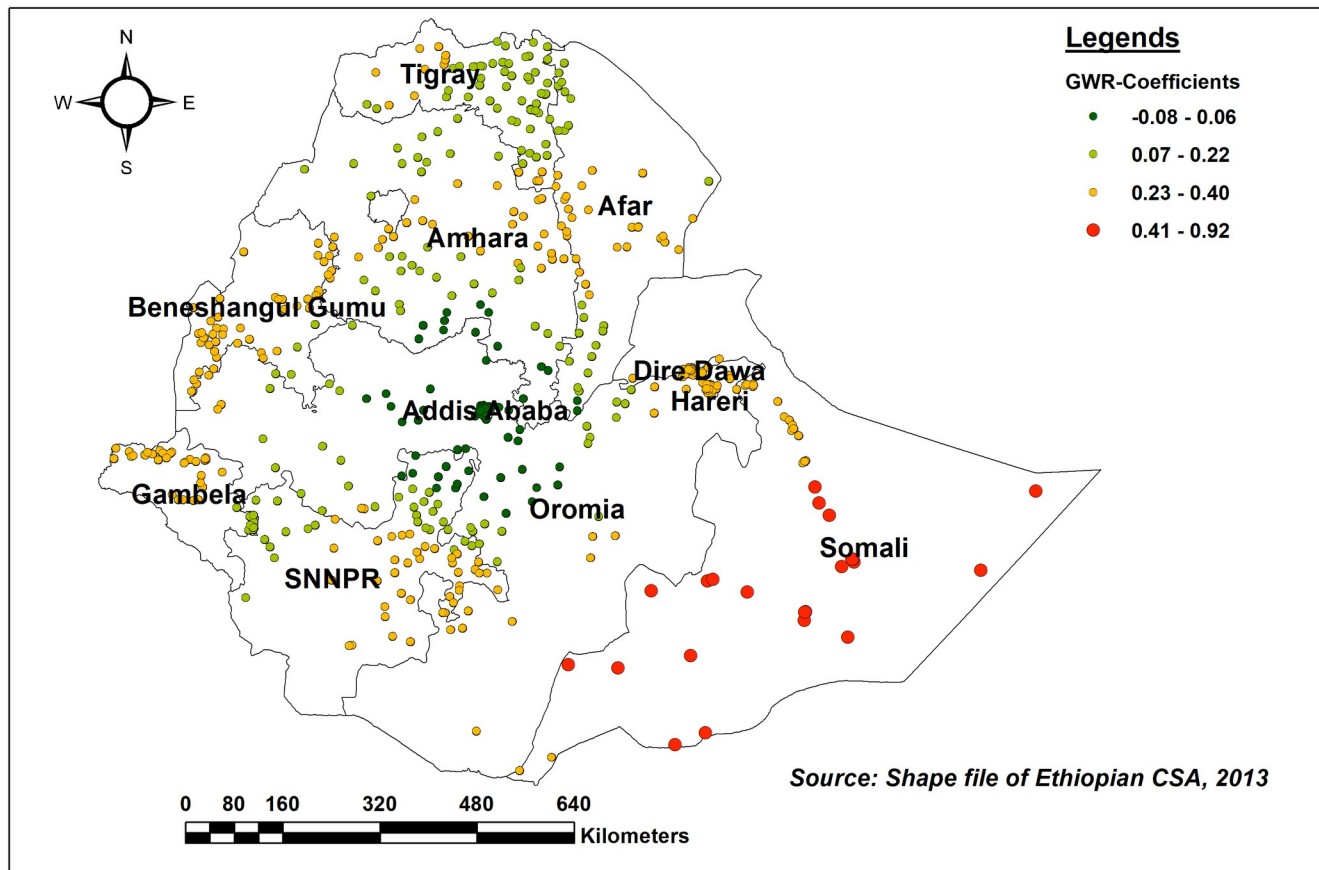

**Fig 13. GWR coefficients of the proportion of women who didn't visit health facilities within the last 12 months for predicting the level of inadequate HIV/AIDS-KAB, EDHS 2016.**

reproductive age. The identified hot spots with relatively poor HIV/AIDS-KAB should be targeted during resource allocation and interventions as well as further research to identify unique local factors and solutions.

**Table 6. Random intercept variances and model fit statistics comparison of multilevel mixed effect logistic regression model.**

| Measures | Model I (null model) | Model II (individual-level factors) | Model III (community-level factors) | Model-IV (full model) |
|---|---|---|---|---|
| **Random effects** | | | | |
| Variance | 1.60 | 0.33 | 0.268 | 0.30 |
| ICC | 0.331 | .144 | 0.144 | 0.088 |
| AIC | 14952.23 | 13584.6 | 14476.6 | 13573.1 |
| MOR | 3.34 | 1.17 | 1.64 | 1.13 |
| PCV | Ref. | 79.3% | 55.6% | 81.25% |
| **Model fitness** | | | | |
| Log-likelihood | -7474.1 | -6762.3 | -7230.3 | -6653.6 |
| Deviance | 14948.2 | 13524.6 | 14460.6 | 13307.2 |

**Table 7. Results of a multilevel multivariable logistic regression to identify the determinants of inadequate HIV/AIDS-KAB among women of reproductive age group, EDHS 2016.**

| Variable categories | Model I (null model) | Model II (individual-level factors) | Model III (community-level factors) | Model-IV (full model) |
|---|---|---|---|---|
| | | AOR(95%CI) | AOR (95%CI) | AOR (95%CI) |
| **Current age** | | | | |
| 15–24 | | Ref. | | Ref. |
| 25–34 | | 0.90(0.74, 1.09) | | 0.92(0.75, 1.11) |
| 35–49 | ' | 0.92(0.72, 1.15) | | 0.95(0.75, 1.19) |
| **Religion** | | | | |
| Orthodox | | Ref. | | Ref. |
| Muslim | | 1.62(1.34, 1.94) | | **1.54(1.27, 1.87)** |
| Protestant | | 1.17(0.96, 1.44) | | 1.15(0.94, 1.42) |
| Catholic | | 2.11(0.77, 5.81) | | 2.05(0.76, 5.53) |
| Others | | 1.63(0.88, 3.01) | | 1.60(0.86, 2.94) |
| **Wealth index combined** | | | | |
| Poorest | | 1.93(1.49, 2.50) | | 1.63(1.21, 2.18) |
| Poorer | | 1.37(1.08, 1.73) | | 1.17(0.88, 1.55) |
| Middle | | 1.50(1.17, 1.92) | | 1.29(0.97, 1.73) |
| Richer | | 1.25(0.99, 1.57) | | 1.08(0.82, 1.42) |
| Richest | | Ref. | | Ref. |
| **Educational status** | | | | |
| No education | | 2.71(2.08, 3.55) | | 2.66(2.04, 3.48) |
| Primary | | 1.66(1.35, 2.04) | | 1.64(1.34, 2.02) |
| Secondary and higher | | Ref. | | Ref. |
| **Parity** | | | | |
| Nulliparous | | Ref. | | Ref. |
| Primiparous | | 1.08(0.87, 1.33) | | 1.07(0.86, 1.32) |
| Multiparous | | 0.84(0.67, 1.05) | | 0.83(0.66, 1.04) |
| Grand multiparous | | 0.91(0.68, 1.21) | | 0.87(0.65, 1.17) |
| **Contraceptive usage** | | | | |
| Yes | | Ref. | | Ref. |
| No | | 1.24(1.08, 1.43) | | 1.23(0.97, 1.42) |
| **Autonomy in decision making** | | | | |
| Low | | 1.83(1.45, 2.30) | | **1.80(1.43, 2.28)** |
| Medium | | 1.31(1.04, 1.64) | | 1.30(0.99, 1.63) |
| High | | Ref. | | Ref. |
| **Ease of distance to seek healthcare** | | | | |
| Big problem | | 1.09(0.92, 1.30) | | 1.07(0.90, 1.28) |
| Not a big problem | | Ref. | | Ref. |
| **Ease of money to seek healthcare** | | | | |
| Big problem | | 1.16(0.98, 1.36) | | 1.07(0.98, 1.38) |
| Not a big problem | | Ref. | | Ref. |
| **Listen to radio** | | | | |
| Not at all | | 1.55(1.02, 2.38) | | **1.56(1.02, 2.39)** |
| Less than once a week | | 0.98(0.65, 1.48) | | 0.99(0.66, 1.49) |
| At least once a week | | Ref. | | Ref. |
| **Watching TV** | | | | |
| Not at all | | 1.63(1.29, 2.05) | | **1.50(1.17, 1.92)** |
| Less than once a week | | 1.31(1.01, 1.70) | | 1.23(0.94, 1.615) |
| At least once a week | | Ref. | | Ref. |

*(Continued)*

**Table 7.** (Continued)

| Variable categories | Model I (null model) | Model II (individual-level factors) | Model III (community-level factors) | Model-IV (full model) |
|---|---|---|---|---|
| | | AOR(95%CI) | AOR (95%CI) | AOR (95%CI) |
| **Reading newspaper** | | | | |
| Not at all | | 1.28(1.05, 1.56) | | 1.19(0.98, 1.58) |
| Less than once a week | | 1.21(0.97, 1.53) | | 1.22(0.97, 1.54) |
| At least once a week | | Ref. | | Ref. |
| **Own mobile phone** | | | | |
| No | | 1.56(1.28, 1.89) | | **1.45(1.27, 1.88)** |
| Yes | | Ref. | | Ref. |
| **Visit health facility within the last 12 months** | | | | |
| Yes | | Ref. | | Ref. |
| No | | 1.49(1.29, 1.73) | | **1.46(1.28, 1.72)** |
| **Residence** | | | | |
| Urban | | | Ref. | Ref. |
| Rural | | | 2.61(1.97, 3.45) | **1.42(1.18, 1.93)** |
| **Regions** | | | | |
| Major central regions | | | 1.03(0.80, 1.33) | 0.97(0.71, 1.15) |
| Peripheral | | | 1.90(1.45, 2.50) | **1.87(1.48, 3.85)** |
| Metropolitans | | | Ref. | Ref. |
| **Community level women illiteracy** | | | | |
| Low | | | Ref. | |
| High | | | 1.73(1.43, 2.11) | 1.26(0.91, 1 42) |
| **Community level poverty** | | | | |
| Low | | | Ref. | Ref. |
| High | | | 1.42(1.14, 1.77) | 1.08(0.88, 1.34) |
| Community level media exposure | | | | |
| Low | | | 1.34(1.08, 1.65) | 0.98(0.77, 1.24) |
| High | | | Ref. | Ref. |

**Key:** Ref.: Reference category; aOR = Adjusted Odds Ratio, * statistically significant at p <0.05, ** statistically significant at p <0.001

## Supporting information

**S1 File. The incremental autocorrelation of inadequate HIV/AIDS comprehensive knowledge among Ethiopian women, EDHS 2016.**
(PDF)

**S2 File. The most likely SaTScan clusters of areas with a high prevalence of inadequate knowledge on HIV transmission among women in Ethiopia, EDHS 2016.**
(DOCX)

## Acknowledgments

We would like to acknowledge the DHS office to access the data based on a reasonable request.

## Author Contributions

**Conceptualization:** Aklilu Habte.

**Data curation:** Aklilu Habte.

**Formal analysis:** Aklilu Habte, Habtamu Mellie Bizuayehu, Yordanos Sisay Asgedom.

**Investigation:** Aklilu Habte.

**Methodology:** Aklilu Habte, Habtamu Mellie Bizuayehu, Yosef Haile, Daniel Niguse Mamo.

**Resources:** Aklilu Habte.

**Software:** Aklilu Habte, Daniel Niguse Mamo.

**Validation:** Aklilu Habte, Yordanos Sisay Asgedom.

**Visualization:** Aklilu Habte, Habtamu Mellie Bizuayehu.

**Writing – original draft:** Aklilu Habte, Habtamu Mellie Bizuayehu.

**Writing – review & editing:** Aklilu Habte, Habtamu Mellie Bizuayehu, Yosef Haile, Daniel Niguse Mamo, Yordanos Sisay Asgedom.

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
