## [Decision Letter · Decision Letter 0]

12 Feb 2024

PONE-D-24-00224Spatial variation and predictors of inadequate HIV/AIDS knowledge among Ethiopian Women: A spatial and multilevel analyses of the 2016 Demographic Health SurveyPLOS ONE

Dear Dr. Habte,

Thank you for submitting your manuscript to PLOS ONE. After careful consideration, we feel that it has merit but does not fully meet PLOS ONE’s publication criteria as it currently stands. Therefore, we invite you to submit a revised version of the manuscript that addresses the points raised during the review process.

We look forward to receiving your revised manuscript.

Kind regards,

Clement Ameh Yaro, Ph.D

Academic Editor

PLOS ONE

2. For studies involving third-party data, we encourage authors to share any data specific to their analyses that they can legally distribute. PLOS recognizes, however, that authors may be using third-party data they do not have the rights to share. When third-party data cannot be publicly shared, authors must provide all information necessary for interested researchers to apply to gain access to the data. (https://journals.plos.org/plosone/s/data-availability#loc-acceptable-data-access-restrictions)

a) A description of the data set and the third-party source

b) If applicable, verification of permission to use the data set

c) Confirmation of whether the authors received any special privileges in accessing the data that other researchers would not have

d) All necessary contact information others would need to apply to gain access to the data

3. We note that Figures 2,4,5,6,7,8,9,10,11 and 12 in your submission contain [map/satellite] images which may be copyrighted. All PLOS content is published under the Creative Commons Attribution License (CC BY 4.0), which means that the manuscript, images, and Supporting Information files will be freely available online, and any third party is permitted to access, download, copy, distribute, and use these materials in any way, even commercially, with proper attribution. For these reasons, we cannot publish previously copyrighted maps or satellite images created using proprietary data, such as Google software (Google Maps, Street View, and Earth). For more information, see our copyright guidelines: http://journals.plos.org/plosone/s/licenses-and-copyright.

a. You may seek permission from the original copyright holder of Figures 2,4,5,6,7,8,9,10,11 and 12 to publish the content specifically under the CC BY 4.0 license. 

Reviewers' comments:

Reviewer's Responses to Questions

**Comments to the Author**

1. Is the manuscript technically sound, and do the data support the conclusions?

Reviewer #1: Partly

Reviewer #2: Yes

Reviewer #3: Partly

Reviewer #4: Yes

2. Has the statistical analysis been performed appropriately and rigorously? 

Reviewer #1: Yes

Reviewer #2: No

Reviewer #3: Yes

Reviewer #4: Yes

3. Have the authors made all data underlying the findings in their manuscript fully available?

Reviewer #1: Yes

Reviewer #2: Yes

Reviewer #3: Yes

Reviewer #4: Yes

4. Is the manuscript presented in an intelligible fashion and written in standard English?

Reviewer #1: No

Reviewer #2: Yes

Reviewer #3: Yes

Reviewer #4: Yes

5. Review Comments to the Author

Reviewer #1: If the authors address all the comments and suggestions provided, the paper has the potential to become significant for publication and beneficial for readers. However, if the recommendations are not adequately addressed, it may be advisable to decline the paper. Ensuring that all comments are thoroughly incorporated can significantly enhance the paper's quality, relevance, and contribution to the field. Therefore, it is essential for the authors to carefully consider and implement the suggested revisions to maximize the paper's impact and value for both the scientific community and readers.

Reviewer #2: Abstract

The abstract section you provided appears to be generally well-written and provides a clear overview of the study's objectives, methods, and findings. However, there are a few areas where the abstract could be improved:

Background: It would be helpful to include more detailed information in the background section regarding the research that have been done on Ethiopian women's awareness of HIV/AIDS. It would be useful to specify the number of studies, their scope, and any important conclusions or limits rather than just saying that "some studies" were undertaken. If your goal is to investigate the gaps in your knowledge, then please be content with that.

Objective: Where are the predictors found by the regression analysis in this case that the author stated, "Consequently, this study attempted to assess the prevalence, geographical variation (Hotspots), spatial predictors, and multilevel correlations of inadequate HIV/AIDS awareness among Ethiopian women"? Because of this, the spatial regression method was used to identify your predictors instead of the multilevel regression analysis. What does little understanding of HIV/AIDS mean in terms of multilayer correlates? The writers must choose the appropriate regression model and have a well-defined target.

Methods: Although the study's use of data from the 2016 Ethiopian Demographic and Health Survey is noted, it would be beneficial to include a brief overview of the survey's methodology and sampling plan. This would provide readers a clearer picture of the sample's representativeness and the source of the data.

Results: The prevalence of inadequate knowledge about HIV/AIDS among Ethiopian women is presented in the abstract; nonetheless, it would be helpful to give a brief overview of the main multilevel and spatial correlates of inadequate knowledge. This would help readers understand the causes of insufficient knowledge and the extent of their impacts.

Which source did you use to get the reference group variables?

Conclusion: Expanding the conclusion section to include more detailed suggestions based on the data would be beneficial. For instance, it might outline the precise treatments or tactics that could be used to close the observed knowledge gaps, rather than just recommending the need for customized health education and awareness campaigns.

Introduction

The introduction section you provided contains valuable information about the global and Ethiopian context of HIV/AIDS and the importance of comprehensive knowledge for prevention efforts. However, there are a few areas where the introduction could be improved:

Structure: There is room for improvement in the introduction's structure. Think about breaking up the section into paragraphs that each focus on a different facet of the subject. The content would flow more easily for readers as a result.

Citations: It would be beneficial to give precise citations for each piece of material, even though you have included some references to back up the claims stated in the introduction. This will guarantee the accuracy of the material provided and allow readers the chance to go further into the research that are cited for further details.

Clarity and Conciseness: The introduction has a few long and complicated sentences that may be challenging to grasp. To make the text easier to understand, try to make the language simpler and divide longer, more complicated statements into shorter, more direct ones.

Research Gap: Although the introduction notes that earlier research on Ethiopian women's comprehensive knowledge of HIV/AIDS was small in scale and scope, it would be helpful to highlight the particular research gap that this study attempts to fill. Indicate exactly how this study is different from the others and what special contributions it will make. In comparison to other earlier studies, the research gap was not often addressed or stated very clearly.

Significance: The study's possible ramifications for community empowerment, policy decisions, and focused interventions are touched upon in passing in the introduction. Explain the wider significance of filling the identified research gap as well as any possible practical ramifications in order to strengthen this section.

The introductory section can be improved to give a more condensed and targeted summary of the background, purpose, and importance of the study by taking these ideas into consideration.

Regarding SDG 3, the references 3 and 4 were incorrect. Kindly make use of the primary source for your information.

Some of the references are outdated (e.g. reference 13 was published since 1997)

Although the author claimed that references 20 and 23–25 were narrow in scope and did not examine the elements at the person and community levels, these references are not as narrow as the author claimed.

There are instances of misspellings, acronyms, and capitalizations throughout the entire document. Therefore, kindly make the necessary revisions.

Methods

First create an independent section about the study setting, and period, before you go to the data and population section.

Data and population

Women who have missing data about their residential area locations via the geographical positioning system (GPS) were excluded from the spatial analysis. How did you do it? Does this meet the proper exclusion criteria for your study over the other criteria? Create an independent section titled “Source and Study Population," and then you might narrate the criteria for your study.

Sampling procedures

Please kindly include more depth information about it for readers. We did not expect to ready your references always. Put them according to the previous studies or the DHS information (for illustration urban versus rural household allocations, and total EAs in Ethiopia were not addressed). Furthermore, the authors did not inform us about data collection tools, however, they put it in the title as a subsection in the methods

References 21, 27, and 28 are not your appropriate evidence. Moreover, the publication data (1998) and the information inside it are quite different with regard to your interest. Kindly revise them.

Measurement of variables of the study

The author should see it again in this section. First, the outcome ascertainment should be based on the guide to DHS, not on the interest of the author. Second, utilizing “mean” on compressive knowledge is not recommended by the guidelines; however, the author did it. Why? Based on what criteria or reference they used, Third, adequate and inadequate categorization should be read again by the author. Fourth, there are also attitude- or practice-related questions in their tool; however, they treated them for knowledge, and this should be taken into consideration. Finally, the author provided a reference for their outcome classification (reference 29); however, this reference was not the right reference for the outcome of interest. Outcome ascertainment is the heart of every research project and should be corrected properly before going to the next step.

Explanatory variables

The author missed some important variables such as working or employment status, ANC, place of delivery, health visits, modern contraceptive utilization (the given contraceptive utilization should be clearly defined rather than using any methods), sex of the given household head, breastfeeding status, and other individual-level factors. Fortunately, all these variables are available in the EDHS 2016 dataset. Furthermore, the author failed to include important community-level factors, such as community women's wealth, community women's education level, community-level ANC coverage, community-level women's mass media exposure, and so on. Kindly include all these and other important variables in your outcome.

Variable selection and categorization should be done in an acceptable manner. For instance, media exposure was better classified into either of the three types (TV, radio, or newspapers). Having a mobile phone or not will not have a great implication in the context of Ethiopia. (Assume your data collection period.)

Data management and statistical analysis

Why used weighting factor v005/1000000, why not other numbers like 1000, or 10000000

If you used GWR, why did you use a multilevel analysis again? or you did not do GWR. The GWR is the regression component for the spatial analysis, and if you applied the GWR, then your probability of coming back again to the multilevel analysis is not feasible. Spatial and multilevel analyses are quite different approaches, and the best regression for you is GWR rather than GLMM.

Use associated factors, or determinants instead of predictors

May the author show us the dataset of outcome variable ascertainments, and the performance of the OLS, and GWR regressions or the do file?

Use the phrase multilevel or mixed effect analysis separately, they are the same thing. Don’t use them as multilevel mixed effect analysis

The outcome variable categorization should be the same for the spatial and regression analyses, either inadequate or adequate knowledge about HIV/AIDS

Use either AIC, or BIC. These two are have different assumptions and working correlation regarding the complexity and parameters of the given model

Although the data has missing values, the author failed to inform how they managed the missing values.

Results

Somalia Vs Somali, should be corrected

The results sections seem discussion due its lot interpretation

Put number before the percentage. For instance, (876 (34.3%)

There is inconsistency of variables utilization throughout the manuscript. Variables from the regression table should be incorporated in the descriptive sections first

Put the PCV, and MOR

Please interpret and talk about the OLS model Diagnosis in your article

Explain the random variation in better please

Use either AOR or aOR, including the abstract section

Discussion

Overall, the discussion section provides a comprehensive analysis of the study's findings and their implications. However, there are a few areas where the discussion could be improved:

Interpretation of Findings: The discussion primarily presents the findings without delving into their underlying causes or providing in-depth interpretations. For example, when discussing the determinants of inadequate comprehensive knowledge, it would be beneficial to explore the mechanisms through which factors such as education, wealth quintile, autonomy, and media exposure influence knowledge levels. Providing a more nuanced understanding of these relationships would enhance the discussion.

Comparison with Other Studies: The discussion includes comparisons with studies conducted in other countries, highlighting differences in knowledge levels. However, it would be helpful to provide more context by discussing potential reasons for the variations observed. What are the specific factors that contribute to higher or lower knowledge levels in different countries? This would allow for a more meaningful comparison and a deeper understanding of the factors influencing knowledge levels.

Implications and Recommendations: While the discussion briefly mentions the need for tailored interventions and policy initiatives, it could benefit from more concrete recommendations based on the study's findings. For instance, how can public health campaigns, education programs, and policy initiatives be designed to address the identified determinants of inadequate knowledge? What specific strategies can be employed to improve access to education, healthcare, and media exposure? Providing practical recommendations would enhance the applicability of the study's findings.

Limitations: The discussion mentions the limitations of the study, such as the cross-sectional nature of the data and the potential for recall and social desirability biases. However, it would be beneficial to discuss these limitations in more detail and consider their impact on the study's findings. Additionally, it would be helpful to suggest avenues for future research that could address these limitations and provide a more robust understanding of HIV/AIDS knowledge among Ethiopian women.

By addressing these suggestions, the discussion section can be improved to provide a more thorough analysis of the study's findings, their implications, and recommendations for future interventions and research.

Conclusion

Try to include the implication of your study briefly.

Reviewer #3: This is very interesting paper as it incorporates different statistical approaches to identify factors associated with inadequate HIV/AIDS knowledge among Ethiopian women

Methodology

1.The prevalence of inadequate HIV/AIDS knowledge among Ethiopian women was reported 48.9%. Considering the cross-sectional nature of the data and the large prevalence of the outcome variable, odds ratio might overestimate the association between the dependent and independent variables. Therefore in my view multilevel Poisson regression analysis with robust variance would be preferable.

2.To show the scale of spatial variation in the relationship between the outcome and predictors, the GWR bandwidth (whether adaptive or fixed) should be reported.

Results

1.In the Satscan analysis, the map with circular windows should be mapped

2.In the interpretation of figure 10, 11 and 12, negative coefficients needs to be interpreted. The authors should also consider demonstrating what positive or negative coefficients mean.

Discussion

1.The GWR analysis, which is the major part of the study was not discussed. Authors should discuss the findings well.

Reviewer #4: I would like to say thank you for considering me to review your paper which is current issue entitled as “Spatial variation and predictors of inadequate HIV/AIDS knowledge among Ethiopian Women: A spatial and multilevel analyses of the 2016 Demographic Health Survey”.

Generally, it is interesting title, information is current issue in the world and the information is so far limited, however the data source is too outdated. Saying this the following massage and question should be addressed.

1.You should revise the manuscript based on Plose one manuscript preparation guideline

2.In your manuscript, you have used abbreviation in title and abstract part. Do you believe this is appropriate and advisable in scientific research journal?

3. Should write the study design clearly you used from your data source?

4.What is the importance of doing geographically weighted regression and logistic regression simultaneously? I think geographically weighted regression is strong enough to generate your evidence?

5.

6. PLOS authors have the option to publish the peer review history of their article (what does this mean?). If published, this will include your full peer review and any attached files.

Reviewer #1: No

Reviewer #2: **Yes: **Bewuketu Terefe

Reviewer #3: No

Reviewer #4: No

---

## [Author Response · Author response to Decision Letter 0]

15 Feb 2024

A point-by-point response to editor and reviewers

Authors’ Response to Academic Editor

Dear: Clement Ameh Yaro, Ph.D, Academic Editor, Plos One

We thank you for a thorough reading and constructive comments and suggestions on our manuscript and for the opportunity to revise and resubmit. We are pleased to submit the revised version of the manuscript titled “Spatial variation and predictors of inadequate HIV/AIDS knowledge among Ethiopian Women: A spatial and multilevel analyses of the 2016 Demographic Health Survey” for your consideration in the special collection of Plos One. The comments of the editors and the reviewers were highly insightful and enabled us to greatly improve the quality of our manuscript. In this revised manuscript we made substantial changes to address your concerns in a point-by-point response. We appreciate your time and look forward to your response and we are very keen to incorporate further comments, if any, for the betterment of the final manuscript.

On the following pages, you will find our responses to the comments and suggestions raised by the esteemed editor and reviewer. 

Sincerely, 

Aklilu Habte(MPH)(corresponding author)

aklilihabte57@gmail.com

Response to Journal requirements

Response: we already prepared the manuscript as per the journal requirement and again we rechecked the compliance towards it during the submission of our revised manuscript.

2. For studies involving third-party data, we encourage authors to share any data specific to their analyses that they can legally distribute. PLOS recognizes, however, that authors may be using third-party data they do not have the rights to share. When third-party data cannot be publicly shared, authors must provide all information necessary for interested researchers to apply to gain access to the data. (https://journals.plos.org/plosone/s/data-availability#loc-acceptable-data-access-restrictions). For any third-party data that the authors cannot legally distribute, they should include the following information in their Data Availability Statement upon submission:

a) A description of the data set and the third-party source

b) If applicable, verification of permission to use the data set

c) Confirmation of whether the authors received any special privileges in accessing the data that other researchers would not have

d) All necessary contact information others would need to apply to gain access to the data

Response: Thank you for your suggestion. However, we already have incorporated in the initial manuscript and now we have highlighted it in the ‘Data availability’ section of the "Revised Manuscript with Track Changes", Lines 580 to 583, Page 24 for your convenience.

3. We note that Figures 2, 4, 5, 6, 7, 8, 9, 10, 11 and 12 in your submission contain map images which may be copyrighted. All PLOS content is published under the Creative Commons Attribution License (CC BY 4.0), which means that the manuscript, images, and Supporting Information files will be freely available online, and any third party is permitted to access, download, copy, distribute, and use these materials in any way, even commercially, with proper attribution. For these reasons, we cannot publish previously copyrighted maps or satellite images created using proprietary data, such as Google software (Google Maps, Street View, and Earth). 

You may seek permission from the original copyright holder of Figures to publish the content specifically under the CC BY 4.0 license. 

Response: We appreciate your concern to assure the ethical issues. However, all the aforementioned figures(2, 4, 5, 6, 7, 8, 9, 10, 11, and 12) in our manuscript are not copyrighted rather they are the result of spatial analysis that we have run in ArcGIS and SaTScan software. The GPS and DHS data that contain Shapefile and other relevant variables were obtained from the DHS office by explaining the objective of the study through online requests. Then, in order to get those figures, we import the relevant data extracted from the 2016 Ethiopian Demographic Health Survey reports and the shapefile of Ethiopia obtained from the 2016 Ethiopian Central Statistical Agency (CSA).To indicate this, we already cited the source of the shapefile alongside each figure. The shape file that we used to construct the figures can be accessed by one of the following links:

1. https://data.humdata.org/dataset/cb58fa1f-687d-4cac-81a7-655ab1efb2d0

2. https://gadm.org/download_country.html

Therefore, the maps presented in our study are not copyrighted rather they were the outputs of our spatial analysis results which are the result of those Shapefiles and projected CVS files in ArcGIS. This is the actual procedure that we employed in our present and earlier studies, as well as other Ethiopian researchers. Again we assure you that the figures presented in our study are not copyrighted but rather our spatial analysis results.

Authors’ Response to Reviewer#1 

General Comment: If the authors address all the comments and suggestions provided, the paper has the potential to become significant for publication and beneficial for readers. However, if the recommendations are not adequately addressed, it may be advisable to decline the paper. Ensuring that all comments are thoroughly incorporated can significantly enhance the paper's quality, relevance, and contribution to the field. Therefore, it is essential for the authors to carefully consider and implement the suggested revisions to maximize the paper's impact and value for both the scientific community and readers.

Response: Thank you for giving us your valuable time to review the manuscript and all your thoughtful comments and suggestions. We got all of them valuable in the way to improve the paper. In the following section, you can get all our responses by highlighting them in the revised version of the manuscript.

Comment 1: I would like to emphasize the need for a clear explanation from the authors regarding the rationale behind conducting both multilevel and geographically weighted regression analyses. Specifically, it is essential to address why the geographically weighted regression (GWR) was chosen and how it demonstrates local statistical significance spatially. If GWR already indicates spatial significance, the inclusion of multilevel logistic regression requires justification to avoid redundancy and ensure the efficiency of the analysis. Clarification on this aspect is strongly warranted to strengthen the paper's coherence and contribute to a better understanding of the analytical approach 

Response: Thank you for your insightful inquiry. Generally, Multilevel analysis and geographically weighted regression (GWR) are two statistical techniques that can be used to examine hierarchical and spatial data, and combining them can offer a more comprehensive understanding of complex spatial relationships. Here's why the integration of multilevel analysis as a support for geographically weighted regression is important:

1. GWR is designed to capture spatial variations in relationships between variables. By incorporating multilevel analysis, we can better account for the nested structure of the data and understand how spatial heterogeneity operates at different levels. It minimizes the prediction variance by giving more weight to nearby sample points with similar values. This helps in obtaining more reliable predictions, especially in areas where the data is sparse.

2. Combining these techniques may lead to improved model performance by accounting for both global and local variations. It helps in avoiding oversimplification of spatial relationships, offering a more nuanced and accurate representation of the data. This can have practical implications for policymakers, as it helps identify areas where interventions may be more effective or where different policies may be needed.

Regarding the second question, why the geographically weighted regression (GWR) was chosen, and how did it demonstrate local statistical significance spatially?

As there are two ways of spatial regression, namely Ordinary least square(Global model) and GWR(local model). GWR is crucial to identifying spatial predictors and that is why we used it. For more clarity we have tried to incorporate the procedures that we followed during spatial regression in the ‘spatial analysis’ section of the "Revised Manuscript with Track Changes" Page 9, Lines 244-254.

Response to specific comments

Comment 1: The methodology section lacks depth as it fails to mention the specific software utilized for spatial analysis, such as the versions of ArcGIS and SaT Scan employed.

Response: we entierly agree with your point of view and we have corrected and highlighted it in the ‘Abstract’ section of the "Revised Manuscript with Track Changes" Page 2, Lines 58-61. Comment 2: The Results section lacks clarification on whether the outcomes represent clustering, identification of hotspot areas, or the results of the SatScan analysis. This ambiguity obscures the interpretation of the findings and their significance. To enhance clarity and provide a comprehensive understanding, it is crucial to explicitly state the nature of the outcomes, particularly regarding clustering, hotspot identification, and the results derived from the SatScan analysis. This clarification will enable readers to better comprehend the implications of the study's findings and their relevance to the research objectives.

Response: we appreciate your meticulous review and we also understand it. We have tried to make the result section more clear by including other findings. The revised statements were highlighted in the ‘Abstract’ section of the "Revised Manuscript with Track Changes" Page 2, Lines 65-69. Comment 3: In conclusion, it is evident that the paper lacks a comprehensive and insightful summary based on the findings presented. The absence of a conclusion section deprives readers of a clear synthesis of the research outcomes and their implications. Therefore, it is imperative to craft a conclusion that effectively encapsulates the key findings, discusses their significance, and provides insights for future research or practical applications

Response: Again, we reconsidered it as per your suggestion and we corrected it. The corrected statements were highlighted in the ‘Abstract’ section of the "Revised Manuscript with Track Changes" Page 2, Lines 77-81. Comment 4: The paper overlooks an essential aspect by failing to mention the data source and provide a direct link to the DHS portal site. Including this information is crucial for transparency and reproducibility, as it enables readers to access the original data for validation or further analysis. Therefore, it is imperative to incorporate a clear statement specifying the data source and providing a hyperlink to the DHS portal site, ensuring that readers have easy access to the dataset used in the study. This addition will enhance the paper's credibility and facilitate future research endeavors by enabling others to replicate or build upon the findings.

Response: thank you for your comment and suggestion. We admit that we ignored the main part and now we incorporated that information in the ‘Study area, period and data source’ section of the "Revised Manuscript with Track Changes" Page 4, Lines 142-144. 

Comment 5: The absence of a map depicting the study area is a notable omission in the paper. Including a map provides essential context for readers to visualize the geographical scope of the study and understand the spatial distribution of the variables under investigation. Incorporating a map showcasing the study area not only enhances the clarity of the research but also enables readers to better interpret the findings in relation to geographic features and boundaries. Therefore, it is recommended to include a detailed map illustrating the study area, its relevant geographical features, and any pertinent spatial data layers. This addition will enrich the paper by providing a visual representation of the research context, thereby enhancing its overall comprehensibility and impact.

Response: thank you for your valuable comment. Accordingly, we have added the administrative map of the country as per your suggestion in Figure 1 and highlighted it in the ‘Study area, period and data source’ section of the "Revised Manuscript with Track Changes" Page 5, Line 148.

Comment 6: The author's focus solely on community-level variables limited to region and residence overlooks other potentially significant factors such as community-level educational status, wealth status, media exposure, and others. These variables play crucial roles in shaping community dynamics and could significantly influence the outcomes under investigation. Therefore, it is essential to address this oversight and consider incorporating a broader range of community-level variables in the analysis. By including additional variables such as educational status, wealth status, and media exposure at the community level, the study can provide a more comprehensive understanding of the factors impacting the outcomes of interest. This expansion of variables will enrich the analysis and enhance the depth of insights derived from the research findings.

Response: we appreciate your meticulous review and insightful suggestions, Accordingly, we included all the above-mentioned community-level factors and highlighted them in Tables 1,2, and 3 of the "Revised Manuscript with Track Changes’’

Comment 7: The absence of incremental autocorrelation analysis in the data analysis section is a notable oversight, particularly concerning autocorrelation analysis. Incremental autocorrelation analysis is crucial for understanding spatial dependence at varying distance thresholds, which can reveal important insights into the clustering patterns within the dataset. Without considering distance thresholds, the autocorrelation analysis may lack precision and potentially overlook significant spatial clustering trends. Therefore, it is imperative to address this deficiency and incorporate incremental autocorrelation analysis into the data analysis methodology

Response: Thank you for your insightful comment and suggestion. Accordingly, we incorporated and highlighted it in the ‘Spatial and incremental autocorrelation’ section of the "Revised Manuscript with Track Changes" Page 8, Lines 209-215. We also reported the findings in the ‘Results’ section, Page 14, Lines 325-329. We added the details as supportive information (S1File)

Comment 8: The Median Odds Ratio (MOR) in the Results section of the multilevel mixed-effect logistic regression analysis is not mentioned, especially considering its importance in understanding the variation between areas of highest and lowest risk. MOR provides valuable insights into the heterogeneity between different geographical areas, highlighting the extent to which individual-level characteristics versus contextual factors contribute to the observed variations in risk. Therefore, it is crucial to address this omission and include MOR analysis in the Results. Response: We appreciate your meticulous review and insightful comment. As per your suggestion, we have added both the narration and the results of MOR and highlighted them in the ‘method’ (Page 10, Lines 275-277) and Result (Page 17, Lines 423-429) of the "Revised Manuscript with Track Changes".

Comment 9: The discussion section appears to lack depth in its interpretation of the findings, particularly in integrating and comparing results from different analytical approaches such as hotspot analysis, SatScan, and others. Each of these methods provides unique insights into spatial patterns and clusters, and their integration can enrich the understanding of the phenomenon under study. Therefore, it is essential to thoroughly discuss and contrast the findings obtained from each method, providing justification for any discrepancies observed

Response: Thank you for your indept

---

## [Decision Letter · Decision Letter 1]

6 Mar 2024

PONE-D-24-00224R1Spatial variation and predictors of inadequate HIV/AIDS knowledge among Ethiopian Women: A spatial and multilevel analyses of the 2016 Demographic Health SurveyPLOS ONE

Dear Dr. Habte,

Thank you for submitting your manuscript to PLOS ONE. After careful consideration, we feel that it has merit but does not fully meet PLOS ONE’s publication criteria as it currently stands. Therefore, we invite you to submit a revised version of the manuscript that addresses the points raised during the review process.

We look forward to receiving your revised manuscript.

Kind regards,

Clement Ameh Yaro, Ph.D

Academic Editor

PLOS ONE

Reviewers' comments:

Reviewer's Responses to Questions

**Comments to the Author**

1. If the authors have adequately addressed your comments raised in a previous round of review and you feel that this manuscript is now acceptable for publication, you may indicate that here to bypass the “Comments to the Author” section, enter your conflict of interest statement in the “Confidential to Editor” section, and submit your "Accept" recommendation.

Reviewer #2: (No Response)

Reviewer #4: All comments have been addressed

2. Is the manuscript technically sound, and do the data support the conclusions?

Reviewer #2: Yes

Reviewer #4: Yes

3. Has the statistical analysis been performed appropriately and rigorously? 

Reviewer #2: No

Reviewer #4: Yes

4. Have the authors made all data underlying the findings in their manuscript fully available?

Reviewer #2: Yes

Reviewer #4: Yes

5. Is the manuscript presented in an intelligible fashion and written in standard English?

Reviewer #2: Yes

Reviewer #4: Yes

6. Review Comments to the Author

Reviewer #2: I beg you to disagree about your outcome ascertainment technique. I have seen the guidelines for DHS statistics, and they do not support your methods of outcome ascertainment.

Reviewer #4: I could not agree more, I have no concern on the issue I raised before. So I believe it can considered for publication

7. PLOS authors have the option to publish the peer review history of their article (what does this mean?). If published, this will include your full peer review and any attached files.

Reviewer #2: No

Reviewer #4: **Yes: **Abel Endawkie

---

## [Author Response · Author response to Decision Letter 1]

7 Mar 2024

A point-by-point response to editor and reviewers

Authors’ Response to Academic Editor

Dear: Clement Ameh Yaro, Ph.D, Academic Editor, Plos One

We thank you for a thorough reading and constructive comments and suggestions on our manuscript and for the opportunity to revise and resubmit. We are pleased to submit the revised version of the manuscript titled “Spatial variation and predictors of comprehensive HIV/AIDS knowledge, attitude and behaviours among Ethiopian Women: A spatial and multilevel analyses of the 2016 Demographic Health Survey” for your consideration in the special collection of Plos One. The comments of the editors and the reviewers were highly insightful and enabled us to greatly improve the quality of our manuscript. In this revised manuscript we made substantial changes to address your concerns in a point-by-point response. We appreciate your time and look forward to your response and we are very keen to incorporate further comments, if any, for the betterment of the final manuscript.

On the following pages, you will find our responses to the comments and suggestions raised by the esteemed reviewer. 

Sincerely, 

Aklilu Habte (MPH), corresponding author

aklilihabte57@gmail.com

Authors’ Response to Reviewer#2

Comment 1: I beg you to disagree about your outcome ascertainment technique. I have seen the guidelines for DHS statistics, and they do not support your methods of outcome ascertainment.

Response: we appreciate your insightful comment and allowing us to look into the outcome assessment in detail, and we entirely agree with your point of view that the way of HIV/AIDS knowledge assessment and the way we did show slight variation. Accordingly, we have changed the name of the outcome variable from merely comprehensive knowledge to comprehensive HIV/AIDS knowledge, attitude, and behaviour (HIV/AIDS-KAB). We have highlighted the correction throughout the revised version of the manuscript. 

We kindly ask you to look into the revised version of the manuscript.

Thank you for your constructive comments and suggestions, which we got as valuable input in the improvement of the manuscript. We received all of them as a valuable contribution to our ongoing work. 

END________________________________________

 THANK YOU!!!

Authors’ response to Reviewer #4

General comment: I have no concern about the issue I raised before. So I believe it can considered for publication

Response: Thank you for your positive feedback by verifying that we responded to all of your comments and providing your delightful suggestion to the editorial office for the publication of our work.

END________________________________________

 THANK YOU!!!

---

## [Decision Letter · Decision Letter 2]

15 Mar 2024

PONE-D-24-00224R2Spatial variation and predictors of comprehensive  HIV/AIDS knowledge, attitude and behaviours among Ethiopian Women: A spatial and multilevel analyses of the 2016 Demographic Health SurveyPLOS ONE

Dear Dr. Habte,

Thank you for submitting your manuscript to PLOS ONE. After careful consideration, we feel that it has merit but does not fully meet PLOS ONE’s publication criteria as it currently stands. Therefore, we invite you to submit a revised version of the manuscript that addresses the points raised during the review process.

We look forward to receiving your revised manuscript.

Kind regards,

Clement Ameh Yaro, Ph.D

Academic Editor

PLOS ONE

Reviewers' comments:

Reviewer's Responses to Questions

**Comments to the Author**

1. If the authors have adequately addressed your comments raised in a previous round of review and you feel that this manuscript is now acceptable for publication, you may indicate that here to bypass the “Comments to the Author” section, enter your conflict of interest statement in the “Confidential to Editor” section, and submit your "Accept" recommendation.

Reviewer #2: (No Response)

2. Is the manuscript technically sound, and do the data support the conclusions?

Reviewer #2: Yes

3. Has the statistical analysis been performed appropriately and rigorously? 

Reviewer #2: No

4. Have the authors made all data underlying the findings in their manuscript fully available?

Reviewer #2: Yes

5. Is the manuscript presented in an intelligible fashion and written in standard English?

Reviewer #2: Yes

6. Review Comments to the Author

Reviewer #2: Review to the authors

Title changed

It is important to ensure consistency between the title of a research article and its content to accurately represent the study's focus and findings. In this case, it seems that the title was changed to incorporate the broader scope of the study, including not only HIV/AIDS knowledge but also attitudes and behaviors among Ethiopian women. However, if the content of the manuscript remains unchanged, including the spatial analysis, outcome variable, sample size, and other aspects, it may create confusion or misalignment between the title and the actual study. This could potentially lead to discrepancies or misunderstandings among readers. In this case the author changed their title without any justification. Please see previous published researches. If you your title has been published with similar outcome, model, and data. You can access them online. At this stage follow the guideline at least. I will not review this again, if you do not follow the guideline.

To address this issue, it is recommended that the authors carefully review the manuscript to ensure that any changes made in the title are appropriately reflected throughout the article. This includes updating relevant sections, such as the introduction, methods, and discussion, to align with the expanded scope of the study as reflected in the new title. Additionally, the authors may want to consider providing a brief explanation or justification within the manuscript, clarifying the reasons for the title change and emphasizing that the content of the study remains unchanged. This can help readers better understand the study's objectives and avoid any confusion arising from the inconsistency between the title and the manuscript's content.

Outcome variable

I repeatedly mentioned to you that the way the authors determined the outcome variable is not based on the guidelines; rather, it was generated based on the authors' own assumptions. Unless the authors used the DHS questionnaires for their study, they should also adhere to the methodology outlined in the guidelines for generating the outcome variable.

For your reference, I have attached the instructions from the guidelines to help simplify the process for you.

Comprehensive knowledge about HIV

Percentage of women and men who know that a healthy looking person can have HIV and reject local misconceptions about transmission or prevention of HIV

Definition

1) Percentage of women and men age 15-49 who know that a healthy looking person can have HIV.

2) Percentage of women and men age 15-49 who know that HIV cannot be transmitted by mosquito bites.

3) Percentage of women and men age 15-49 who know that HIV cannot be transmitted by supernatural means.

4) Percentage of women and men age 15-49 who know that a person cannot become infected by sharing food with a person who has HIV.

5) Percentage of women and men age 15-49 who say that a healthy-looking person can have HIV and who reject the two most common local misconceptions.

Indicators 2, 3 and 4 are surveys-specific and may refer to other local misconceptions.

Coverage:

Population base: Women and men age 15-49

Time period: Current status at time of survey

Numerators:

Number of women (or men) who indicate that they: 1) Know that a healthy-looking person can have HIV (women: v756 = 1; men: mv756 = 1)

2) Know that HIV cannot be transmitted by mosquito bites (women: v754jp = 0; men: mv754jp = 0)

3) Know that HIV cannot be transmitted by supernatural means (women: v823 = 0; men: mv823 = 0)

4) Know that a person cannot become infected by sharing food with a person who has HIV (women: v754wp = 0; men: mv754wp = 0)

5) Know that a healthy-looking person can have HIV (see Numerator 1 above) and reject the two most common local misconceptions about HIV transmission or prevention (these two most common misconceptions are footnoted in DHS 7 table 13.2). Survey specific but typically two of the following three (see Calculation below): a) HIV cannot be transmitted by mosquito bites (women: v754jp = 0; men: mv754jp = 0).

b) HIV cannot be transmitted by supernatural means (women: v823 = 0; men: mv823 = 0).

c) A person cannot become infected by sharing food with a person who has HIV (women: v754wp = 0; men: mv754wp = 0)

7. PLOS authors have the option to publish the peer review history of their article (what does this mean?). If published, this will include your full peer review and any attached files.

Reviewer #2: No

---

## [Author Response · Author response to Decision Letter 2]

16 Mar 2024

A point-by-point response to editor and reviewers

Dear: Clement Ameh Yaro, Ph.D, Academic Editor, Plos One

We thank you for a thorough reading and constructive comments and suggestions on our manuscript and for the opportunity to revise and resubmit. We are pleased to submit the revised version of the manuscript titled “Spatial Variation and Predictors of Composite Index of HIV/AIDS Knowledge, attitude and Behaviours among Ethiopian Women: A Spatial and Multilevel Analyses of the 2016 Demographic Health Survey” for your consideration in the special collection of Plos One. The comments of the editors and the reviewers were highly insightful and enabled us to greatly improve the quality of our manuscript. In this revised manuscript we made substantial changes to address your concerns in a point-by-point response. We appreciate your time and look forward to your response and we are very keen to incorporate further comments, if any, for the betterment of the final manuscript.

On the following pages, you will find our responses to the comments and suggestions raised by the esteemed reviewer. 

Sincerely, 

Aklilu Habte (MPH), corresponding author

aklilihabte57@gmail.com

Authors’ Response to Reviewer#2

Comment 1: It is important to ensure consistency between the title of a research article and its content to accurately represent the study's focus and findings. In this case, it seems that the title was changed to incorporate the broader scope of the study, including not only HIV/AIDS knowledge but also attitudes and behaviors among Ethiopian women. However, if the content of the manuscript remains unchanged, including the spatial analysis, outcome variable, sample size, and other aspects, it may create confusion or misalignment between the title and the actual study. This could potentially lead to discrepancies or misunderstandings among readers. In this case the author changed their title without any justification. Please see previous published researches. If you your title has been published with similar outcome, model, and data. You can access them online. At this stage follow the guideline at least. 

Response: We apologize for the inconvenience. We intended to measure the outcome variable more comprehensively. Of course, we understand that DHS assesses comprehensive HIV/AIDS knowledge using the five elements you listed. Because there has been previous research in this area, we attempted to measure comprehensive HIV/AIDS knowledge, attitude, and behavior by incorporating six additional parameters to keep the current study more novel. DHS also has some space to measure the KAB domain with the items we utilize. We kindly ask you to look into the following links:

1. https://dhsprogram.com/publications/publication-fa56-further-analysis.cfm

2. https://dhsprogram.com/topics/HIV-Corner/hiv-kab/

We kindly ask you to understand us we prefer to increase the number of items to make the study more comprehensive and novel ( there was a dearth of studies that measured the KAB domain in this way). We also want to assure you that we made a slight change in the title to address your concerns and we tried to shape all sections of the manuscript accordingly and we have highlighted them throughout the ‘Revised manuscript with track changes’

Based on the reasons that we mentioned, now we are eager to accept your esteemed decision on the final fate of the manuscript. We highly appreciate your time and effort in reviewing the manuscript. 

Thank you for your constructive comments and suggestions, which we got as valuable input in the improvement of the manuscript. We received all of them as a valuable contribution to our ongoing work. 

END________________________________________

 THANK YOU!!!

---

## [Editor Report · Decision Letter 3]

22 May 2024

Spatial variation and predictors of composite index of HIV/AIDS knowledge, attitude and behaviours among Ethiopian Women: A spatial and multilevel analyses of the 2016 Demographic Health Survey

PONE-D-24-00224R3

Dear Dr. Habte,

We’re pleased to inform you that your manuscript has been judged scientifically suitable for publication and will be formally accepted for publication once it meets all outstanding technical requirements.

Kind regards,

Clement Ameh Yaro, Ph.D

Academic Editor

PLOS ONE
---

## [Editor Report · Acceptance letter]

24 May 2024

PONE-D-24-00224R3 

PLOS ONE

Dear Dr. Habte, 

I'm pleased to inform you that your manuscript has been deemed suitable for publication in PLOS ONE. Congratulations! Your manuscript is now being handed over to our production team.

Kind regards, 

on behalf of

Dr. Clement Ameh Yaro 

Academic Editor

PLOS ONE